# Structural insights into ubiquitin recognition and Ufd1 interaction of Npl4

Yusuke Sato[1,2,3,5,7], Hikaru Tsuchiya[4,7], Atsushi Yamagata[1,2,3,6], Kei Okatsu[1,2], Keiji Tanaka[4], Yasushi Saeki [4]* & Shuya Fukai [1,2,3]*

Npl4 is likely to be the most upstream factor recognizing Lys48-linked polyubiquitylated substrates in the proteasomal degradation pathway in yeast. Along with Ufd1, Npl4 forms a heterodimer (UN), and functions as a cofactor for the Cdc48 ATPase. Here, we report the crystal structures of yeast Npl4 in complex with Lys48-linked diubiquitin and with the Npl4-binding motif of Ufd1. The distal and proximal ubiquitin moieties of Lys48-linked diubiquitin primarily interact with the C-terminal helix and N-terminal loop of the Npl4 C-terminal domain (CTD), respectively. Mutational analysis suggests that the CTD contributes to linkage selectivity and initial binding of ubiquitin chains. Ufd1 occupies a hydrophobic groove of the Mpr1/Pad1 N-terminal (MPN) domain of Npl4, which corresponds to the catalytic groove of the MPN domain of JAB1/MPN/Mov34 metalloenzyme (JAMM)-family deubiquitylating enzyme. This study provides important structural insights into the polyubiquitin chain recognition by the Cdc48–UN complex and its assembly.

[1] Institute for Quantitative Biosciences, The University of Tokyo, Tokyo 113-0032, Japan. [2] Synchrotron Radiation Research Organization, The University of Tokyo, Tokyo 113-0032, Japan. [3] Department of Computational Biology and Medical Sciences, Graduate School of Frontier Sciences, The University of Tokyo, Chiba 277-8562, Japan. [4] Laboratory of Protein Metabolism, Tokyo Metropolitan Institute of Medical Science, Tokyo 156-8506, Japan. [5] Present address: Center for Research on Green Sustainable Chemistry, Tottori University, Tottori 680-8582, Japan. [6] Present address: RIKEN Center for Biosystems Dynamics Research, Kanagawa 230-0045, Japan. [7] These authors contributed equally: Yusuke Sato, Hikaru Tsuchiya. *email: saeki-ys@igakuken.or.jp; fukai@iam.u-tokyo.ac.jp

Ubiquitin (Ub) is an essential posttranslational modifier conserved from yeast to mammals[1–3]. Typically, the C-terminal glycine residue of Ub is covalently attached to lysine residues of substrate proteins. Ub itself is also a substrate for ubiquitylation, and forms a covalently linked Ub chain. The Ub chain linked via Lys48 (K48 chain) is the primary targeting signal for proteasomal degradation[1–5]. Polyubiquitylated proteins in isolation can be direct targets of the proteasome, whereas those embedded into membranes or assembled into multisubunit complexes need to be extracted or segregated by the conserved AAA-family ATPase Cdc48 (yeast) or p97/VCP (metazoan) with the aid of its cofactor complex, Ufd1–Npl4 (UN), prior to the proteasomal degradation[6–11]. The cellular function or localization of the Cdc48/p97 ATPase is controlled by many cofactors. The UN heterodimer is one of the best-characterized cofactors[12–15].

The Cdc48/p97 ATPase consists of an N-terminal (N) domain and two ATPase domains (D1 and D2). D1 and D2 form a hexameric double ring with a central pore[16]. The conformation of the N domain relative to the D1-D2 ring is coupled to the nucleotide state of D1; the N domain is located above the D1 ring in the ATP state (up conformation), and is coplanar with the D1 ring in the ADP state (down conformation)[17–21]. In addition, relative rotations between the D1 and D2 rings occur upon binding of ATP to D2. Although understanding the mechanism of the substrate translocation accompanied with the Ub chain binding and ATP hydrolysis has been a long-standing problem, recent structural and functional studies provided important mechanistic insights into reacting steps of the Cdc48/p97–UN complex[22]. The UN heterodimer captures the substrate-attached Ub chain at the initial step[15].

Npl4 is responsible for recognition of the substrate-attached Ub chain, and thus likely mediates a Ub chain-associated reacting step[15]. The mammalian Npl4 (also known as NPLOC4) consists of a Ubx-like (UBXL) domain, zinc-finger domain (previously designated as zf-Npl4), Mpr1/Pad1 N-terminal (MPN) domain, C-terminal domain (CTD), and NZF domain. The mammalian Npl4 binds to Ub chains without linkage specificity through the NZF domain. On the other hand, yeast Npl4 lacks the NZF domain, and binds specifically to K48 chains via zf-Npl4-MPN-CTD[15,23,24]. The MPN domain of Npl4 topologically resembles the catalytic domain of the JAB1/MPN/Mov34 metalloenzyme (JAMM)-family deubiquitylating enzyme (DUB)[19], which accommodates the C-terminal tail of Ub. Therefore, the groove corresponding to the catalytic site of JAMM DUBs[25,26] has been proposed as a potential Ub-binding site of Npl4. However, a recent cryo-EM analysis of the substrate-engaged Cdc48–UN complex at 3.9 Å resolution has shown that the K48 chain on the substrate does not interact with the groove of yNpl4 MPN but with CTD and other regions of MPN[22]. Nevertheless, due to the low local resolution, the density for Ufd1 was not interpreted with the atomic model and the interaction between Npl4 and the two folded Ub moieties has not been characterized in detail[22].

Here, we present the crystal structures of yeast Npl4 (zf-Npl4-MPN-CTD) in complex with Lys48-linked diubiquitin (K48-Ub$_2$) and with Ufd1 at 2.55 and 1.58 Å resolutions, respectively. The crystal structures reveal that the distal and proximal Ub (Ub$^{dist}$ and Ub$^{prox}$, respectively) moieties of K48-Ub$_2$ interact with the C-terminal helix and N-terminal loop (N loop) of Npl4 CTD, respectively, and that Ufd1 binds to the groove of Npl4 corresponding to the catalytic groove of JAMM. In vitro and in vivo mutational analyses confirm that the Npl4 MPN groove is required for binding to Ufd1 but not to K48 chains. Also, Npl4 mutants that are designed to disrupt the Ub binding show that the interaction between the Npl4 C-terminal helix and Ub is critical for the Ub chain binding and stimulation of the ATPase activity of the Cdc48–UN complex. Moreover, we reveal that mutations on the proximal Ub-binding site of Npl4 affect its linkage-specificity. Our crystallographic and biochemical studies provide essentially important insights to understand the reaction mechanism of the Cdc48/p97–UN complex.

## Results

**Structure of Npl4 in complex with K48-Ub$_2$.** The region containing residues 113–580 of *Saccharomyces cerevisiae* Npl4 (yNpl4$^{113–580}$, with the prefix "y" indicating *Saccharomyces cerevisiae* proteins) specifically recognizes K48 chains in vitro (Fig. 1)[15]. The triple E123A K124A E125A mutation was introduced to reduce excess surface conformational entropy[27]. The yNpl4$^{113–580}$ (E123A K124A E125A) protein yielded high-quality crystals, and its structure was determined at 1.72 Å resolution by the single-wavelength anomalous diffraction (SAD) method using the zinc edge (Table 1). We also attempted to crystallize yNpl4$^{113–580}$ in complex with K48 chains. Although K48-Ub$_2$, K48-Ub$_3$, K48-Ub$_4$, and K48-Ub$_5$ were tested for crystallization of the complex, only K48-Ub$_2$ was successfully co-crystallized with yNpl4$^{113–580}$. Eventually, we determined the crystal structure of yNpl4$^{113–580}$ in complex with selenomethionine (SeMet)-labeled K48-Ub$_2$ at 2.55 Å resolution (Fig. 2a and Table 1). The structure was determined by the molecular replacement method using yNpl4$^{113–580}$ alone as the search model. Although molecular replacement using Ub (PDB 1UBQ [https://doi.org/10.2210/pdb1ubq/pdb])[28] as the search model was unsuccessful, we found residual electron density corresponding to K48-Ub$_2$ and manually built the model of K48-Ub$_2$. The final model contains one yNpl4$^{113–580}$–K48-Ub$_2$ complex and one isolated yNpl4$^{113–580}$ molecule in the asymmetric unit. We here note that the electron density of K48-Ub$_2$ is weak, especially of Ub$^{prox}$ (Supplementary Fig. 1a). The electron density of the yNpl4-interacting part of Ub$^{prox}$ is observed, whereas the solvent exposed part of Ub$^{prox}$ is obscured (Supplementary Fig. 1a, b). To confirm the positions of Ub$^{dist}$ and Ub$^{prox}$, we replaced Pro19 Val26 or Ile30 of Ub with SeMet and calculated the anomalous difference Fourier map in the yNpl4$^{113–580}$–K48-Ub$_2$ complex (Supplementary Fig. 1c). Although some signals derived from SeMet were indistinguishable or not detected, we detected the signals derived from SeMet1, SeMet19, and SeMet26 of the Ub$^{dist}$ and SeMet1, SeMet26, and SeMet30 of the Ub$^{prox}$.

The yNpl4$^{113–580}$ structure consists of three subdomains, zf-Npl4, MPN, and CTD, similarly to the previously reported structures of Npl4[19,22]. The yNpl4 MPN subdomain consists of the MPN core and two insertions, Ins-1 (residues 287–317) and Ins-2 (residues 403–462), similarly to other MPN DUBs. The Ub$^{dist}$ moiety contacts the CTD subdomain of yNpl4 with a buried surface area of 460 Å$^2$ (Fig. 2b). The Ub$^{prox}$ moiety contacts the CTD and MPN subdomains of yNpl4 with a buried surface area of 468 Å$^2$. Although the electron density corresponding to the C-terminal five residues (Arg-Leu-Arg-Gly-Gly) of Ub$^{dist}$ and the Lys48 side chain of Ub$^{prox}$ are invisible (Fig. 2c), the Cα–Cα distance between Leu71 of Ub$^{dist}$ and Lys48 of Ub$^{prox}$ (11 Å) is within the range of those in the previously reported crystal structures of K48 chains in complex with their specific effectors (SARS PLpro[29], AIRAPL[30], and Mindy-1[31]; 9–20 Å). Furthermore, we examined the yNpl4–K48-Ub$_2$ crystals by SDS-PAGE (Supplementary Fig. 1d), and confirmed that the linkage of K48-Ub$_2$ was retained in the crystals. Therefore, we concluded that the isopeptide bond connects Ub$^{dist}$ and Ub$^{prox}$ in the present yNpl4–K48-Ub$_2$ structure. No large conformational difference was observed between yNpl4$^{113–580}$ alone and yNpl4$^{113–580}$–K48-Ub$_2$, except for the region comprising residues 290–300 in Ins-1, whose conformation is constrained by crystal packing (Supplementary Fig. 2a).

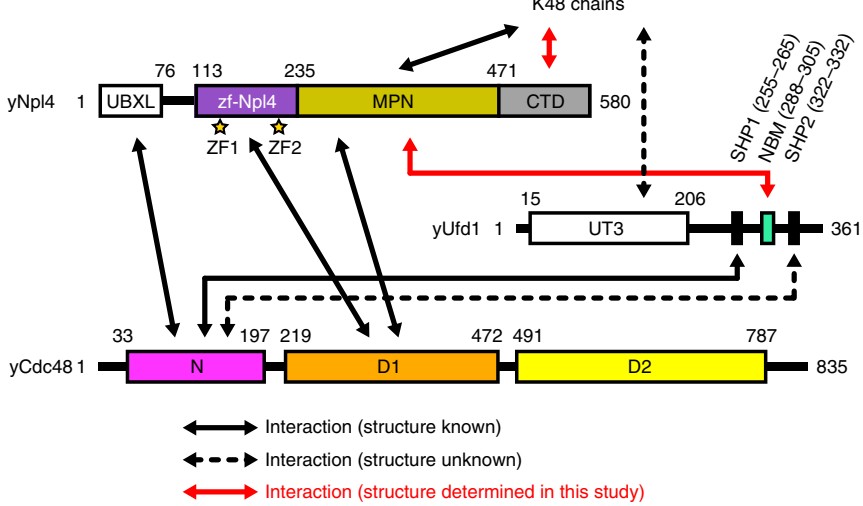

**Fig. 1 Domain compositions of *S. cerevisiae* Npl4, Ufd1, and Cdc48.** The zf-Npl4, MPN, and CTD subdomains of yNpl4 are purple, dark yellow, and gray, respectively. ZF1 and ZF2 of yNpl4 are indicated by yellow stars. The NBM region of Ufd1 is turquoise. The N, D1, and D2 domains of Cdc48 are magenta, orange, and yellow, respectively.

**Table 1 Data collection and refinement statistics.**

| | yNpl4[113-580] (E123A K124A E125A) | yNpl4[113-580]–K48-Ub₂ | yNpl4[113-580] (E123A K124A E125A)–yUfd1[288-305] |
|---|---|---|---|
| Data collection | | | |
| Space group | $P2_12_12_1$ | $P2_1$ | $P2_12_12_1$ |
| Cell dimensions | | | |
| $a, b, c$ (Å) | 76.0, 82.9, 92.1 | 86.4, 103.1, 99.6 | 74.0, 82.5, 94.1 |
| $\alpha, \beta, \gamma$ (°) | 90.0, 90.0, 90.0 | 90.0, 100.4, 90.0 | 90.0, 90.0, 90.0 |
| Resolution (Å) | 50-1.72 (1.75-1.72) | 50.0-2.55 (2.58-2.55) | 50.0-1.58 (1.61-1.58) |
| $R_{sym}$ | 0.089 (0.817) | 0.252 (1.307) | 0.071 (1.449) |
| $R_{pim}$ | 0.018 (0.227) | 0.056 (0.521) | 0.020 (0.512) |
| $I/\sigma(I)$ | 32.5 (1.26) | 10.3 (0.831) | 25.6 (0.554) |
| Completeness (%) | 93.3 (64.0) | 99.1 (91.3) | 96.7 (70.8) |
| CC(1/2) | 0.998 (0.712) | 0.985 (0.504) | 1.000 (0.496) |
| Redundancy | 22.2 (11.2) | 19.3 (6.1) | 12.4 (7.5) |
| Refinement | | | |
| Resolution (Å) | 1.72 | 2.55 | 1.58 |
| No. reflections | | | |
| $R_{work}/R_{free}$ | 16.3/19.3 | 19.1/22.6 | 17.8/20.6 |
| No. atoms | | | |
| Protein | 3709 | 8521 | 3883 |
| Ligand/Ion | 56 | 425 | 68 |
| Water | 406 | 170 | 390 |
| $B$ factors (Å²) | | | |
| Protein | 37.9 | 57.6 | 33.5 |
| Ligand/ion | 63.2 | 74.6 | 59.4 |
| Water | 46.6 | 41.8 | 42.5 |
| R.m.s. deviations | | | |
| Bond lengths (Å) | 0.010 | 0.010 | 0.006 |
| Bond angles (°) | 1.198 | 1.215 | 0.984 |
| Ramachandran plot | | | |
| Favored (%) | 97.8 | 97.9 | 98.5 |
| Outliers (%) | 0.0 | 0.0 | 0.0 |

Values in parentheses are for highest-resolution shell

**The C-terminal helix of Npl4 plays a key role in Ub binding.** Ub[dist] of K48-Ub₂ interacts primarily with the C-terminal helix of yNpl4 CTD (Fig. 2b). Met574 and Ile575 form a hydrophobic surface to interact with the Ile44-centered hydrophobic patch of Ub[dist]. This hydrophobic interaction is further stabilized by hydrogen bonds between Thr571 of yNpl4 and the main-chain NH groups of Ala46 and Gly47 in Ub[dist]. Adjacent to the C-terminal helix-mediated interactions, Tyr501 and Ile538 of yNpl4 project into a hydrophobic pocket formed by Leu8, His68, and Val70 of Ub[dist]. To assess the roles of these interactions in the Ub chain recognition by Npl4, we examined the binding of the T571A, M574A, M574Q, and I575A mutants of yNpl4 to K48 chains by surface-plasmon resonance (SPR) spectroscopy (Table 2). The SPR analysis of mutant yNpl4 was performed

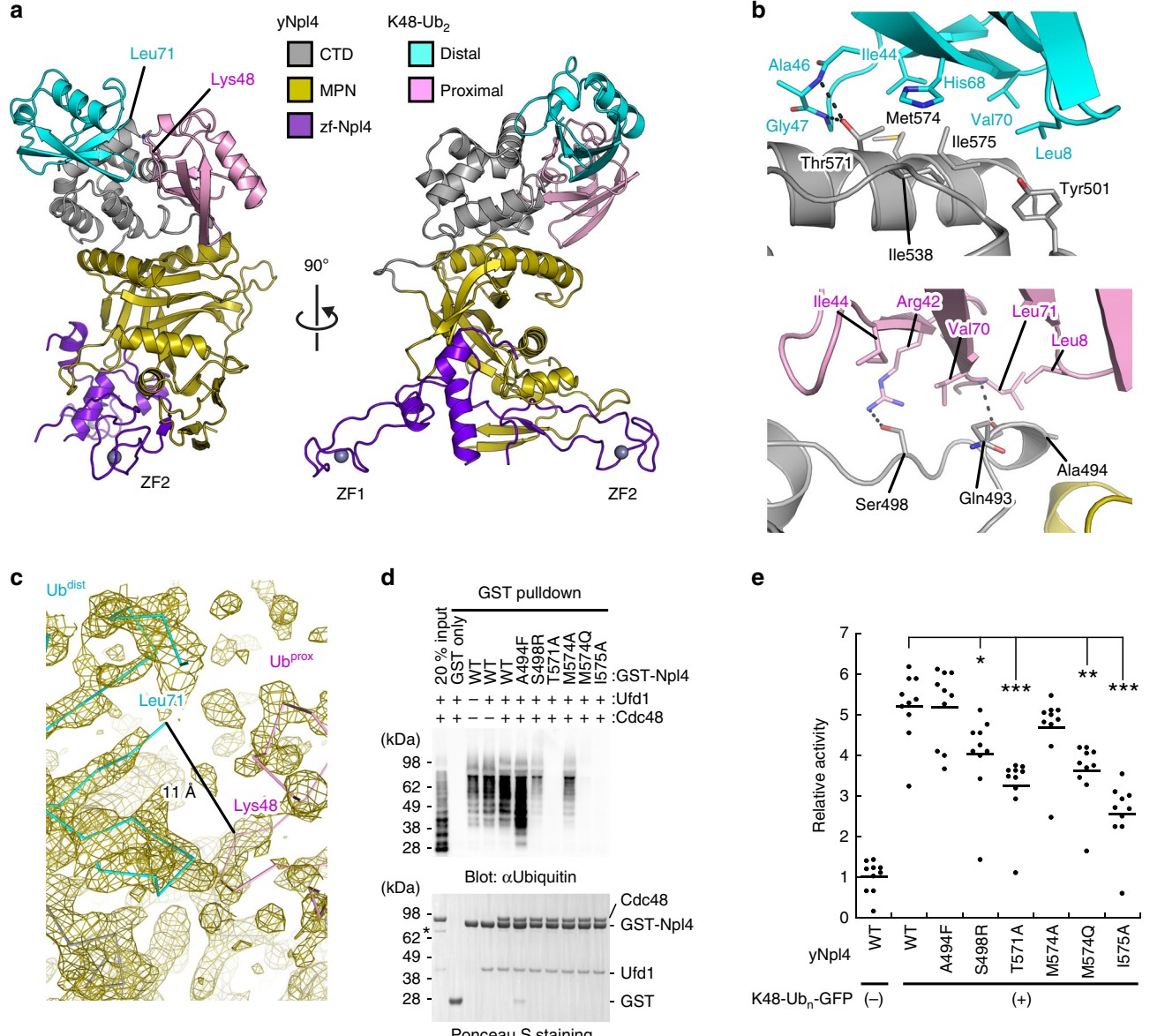

**Fig. 2 Crystal structure of yNpl4 in complex with K48-Ub$_2$. a** Overall structure of the yNpl4–K48-Ub$_2$ complex in two orientations. **b** Close-up view of the interactions between yNpl4 and K48-Ub$_2$. Hydrogen bonds are shown as black dotted lines. **c** Close-up view of the area around the isopeptide linkage of K48-Ub$_2$. A $2F_o$–$F_c$ composite omit map is shown as an olive mesh contoured at 1 $\sigma$ level. **d** Analysis of the binding between K48 chains and the Cdc48-UN complex containing wild-type or mutant GST-yNpl4 by pulldown assays. The bound K48 chains were detected by immunoblotting with anti-Ub antibody (upper panel). Blot membranes were stained with Ponceau S (lower panel). In all, 20% input means 20% of the volume of the sample (K48 chain, Cdc48, and yUfd1) that was mixed with the GST-yNpl4-bound glutathione resin. Asterisks indicate contamination. This experiment was repeated with distinct samples (Supplementary Fig. 3a). Source data are provided as a Source Data file. **e** ATP hydrolysis rates of the Cdc48–UN complex containing wild-type or mutant yNpl4 with or without K48-Ub$_n$-GFP. The rates were normalized to the average of the ATP hydrolysis rates of the wild-type Cdc48-UN complex without K48-Ub$_n$-GFP. The line represents the mean of the rates after normalization (mean values; $n = 10$ independent experiments; *$P < 0.05$; **$P < 0.01$; ***$P < 0.001$ from Tukey's test).

using K48-Ub$_4$ because the affinity for K48-Ub$_2$ was too low to be analyzed. Longer Ub chains bind to their effectors with higher affinity in general[30,32,33]. The T571A mutation of yNpl4[113–580] decreased the affinity for K48-Ub$_4$ to 5.0% of the wild-type affinity (Table 2). The I575A mutation of yNpl4[113–580] decreased the affinity to an unmeasurable level. The M574A mutation of yNpl4[113–580] had a weak effect and decreased the affinity for K48-Ub$_4$ to 43% of the wild-type affinity. On the other hand, The M574Q mutation had a greater effect than the M574A mutation, and decreased the affinity for K48-Ub$_4$ to 9.1% of the wild-type affinity. This is likely because the hydrophilic side chain of Gln

inhibits the hydrophobic interaction between yNpl4[113–580] and Ub$^{dist}$. We further analyzed the affinities between yNpl4[113–580] mutants and K63- or M1-Ub$_4$. The T571A, M574A, M574Q, or I575A mutation decreased the affinity for K63- or M1-Ub$_4$ to an unmeasurable level (Table 2). These results suggest that the yNpl4–Ub$^{dist}$ interface primarily contributes to the affinity for Ub chains rather than the linkage specificity.

Next, the binding between the Cdc48–UN complex and K48 chains was analyzed by GST pulldown assays. GST-yNpl4 mutants were immobilized to glutathione beads and incubated with Cdc48, yUfd1, and K48 chains. After washing, the bound

**Table 2 Binding affinity of Npl4 for K48, K63, or M1 chains.**

| yNpl4 | $K_d$ (μM) | | | Linkage specificity | |
|---|---|---|---|---|---|
| | K48-Ub$_2$ | K63-Ub$_2$ | M1-Ub$_2$ | K48/K63 | K48/M1 |
| WT #1[a] | 42.6 ± 0.1 | 268 ± 1 | (333 ± 1) | 6 | – |
| WT #2[a] | 55.3 ± 0.1 | 327 ± 1 | (420 ± 2) | 6 | – |
| **yNpl4** | **K48-Ub$_4$** | **K63-Ub$_4$** | **M1-Ub$_4$** | **K48/K63** | **K48/M1** |
| WT #1[a] | 6.75 ± 0.11 | 75.5 ± 0.1 | 103 ± 5 | 11 | 15 |
| WT #2[a] | 8.13 ± 0.23 | 88.0 ± 1.3 | 139 ± 3 | 11 | 17 |
| Ub$^{dist}$ recognition region | | | | | |
| T571A | 134 ± 5 | (574 ± 11) | (923 ± 30) | – | – |
| M574A | 15.6 ± 0.2 | (181 ± 3) | (291 ± 5) | – | – |
| M574Q | 73.8 ± 2.3 | (377 ± 7) | (634 ± 13) | – | – |
| I575A | (206 ± 8) | (517 ± 12) | (837 ± 24) | – | – |
| Ub$^{prox}$ recognition region | | | | | |
| A494F | 3.04 ± 0.06 | 62.5 ± 0.1 | 84.0 ± 4.3 | 21 | 28 |
| S498L | 10.4 ± 0.2 | 71.2 ± 0.1 | 101 ± 5 | 7 | 10 |
| S498R | 18.8 ± 1.3 | 90.9 ± 1.6 | 136 ± 3 | 5 | 7 |
| yUfd1–yNpl4 | | | | | |
| WT | 7.61 ± 0.16 | 103 ± 2 | (151 ± 5) | 16 | – |
| **hNpl4** | **K48-Ub$_4$** | | | | |
| WT | 2.21 ± 0.12 | | | | |
| T551A | 3.78 ± 0.14 | | | | |
| Q554M | 1.11 ± 0.09 | | | | |

Data are presented as mean ± standard deviation; $n = 3$ independent experiments
$K_d$ values in parentheses are above half of the upper limit of the substrate concentration used in the experiment, and may be underestimated
[a]The assays of wild-type yNpl4 were performed twice using distinct samples with similar results

proteins were eluted with LDS-loading buffer and analyzed by immunoblotting with anti-Ub antibody (Fig. 2d and Supplementary Fig. 3a). In this assay, yUfd1 or Cdc48 had a little effect on the binding to K48 chains, and the UN or Cdc48–UN complex formation was not affected by the mutations of yNpl4 examined in this study. On the other hands, the T571A, M574Q, or I575A mutation, which had a severe effect on the K48-Ub$_4$ binding, completely abolished the binding of Cdc48–UN for K48 chains. The M574A mutation, which had a mild effect on the affinity for K48-Ub$_4$, did not completely eliminate the binding. Similar GST pulldown assays using GST-yUfd1 instead of GST-yNpl4 showed consistent results (Supplementary Fig. 3b). To further assess the functional significance of the yNpl4–Ub$^{dist}$ interaction in vivo, mutant yNpl4-3xFLAG was expressed under its own promoter in npl4Δ cells. yNpl4-3xFLAG was immunoprecipitated with anti-FLAG antibody, and co-immunoprecipitation of Lys48-linked polyubiquitylated proteins were analyzed with anti-K48 chain antibody (Supplementary Fig. 3c). In agreement with the result of the in vitro binding assay, the T571A, M574Q, or I575A mutation decreased the amount of co-immunoprecipitated Lys48-linked polyUb conjugates, although these mutations had little effect on Cdc48 binding.

Previous studies showed that Ufd1 binds to K48 chains through its UT3 domain[24,34]. However, our pulldown analysis failed to detect the binding of yUfd1 to K48 chains (Supplementary Fig. 3b). We next analyzed the binding between the GST-yNpl4 and K48 chains on increasing concentrations of yUfd1 by GST pulldown assay but did not observe substantial effects (Supplementary Fig. 3d). SPR analysis using the purified yUfd1–yNpl4$^{113-580}$ complex and K48-Ub$_4$ also showed that the presence of yUfd1 has little effect on the affinity of yNpl4 for K48-Ub$_4$ (Table 2). Although the $K_d$ value of yUfd1 for polyUb has not been estimated, that for monoUb was estimated to be within the range of 1–2 mM[34]. This value is much higher than the $K_d$ value of yNpl4 for K48-Ub$_2$ or K48-Ub$_4$ (Table 2), suggesting that the affinity of yUfd1 for K48 chains is much lower than that for yNpl4. Collectively,

these in vitro and in vivo results suggest that the C-terminal helix of yNpl4 is the key determinant for Ub chain recognition by the Cdc48–UN complex.

**The N loop of Npl4 CTD contributes to K48 chain specificity.** Ub$^{prox}$ of K48-Ub$_2$ mainly interacts with the N loop of yNpl4 CTD (residues 493–501) (Fig. 2b). The N loop of yNpl4 CTD interacts with the hydrophobic patch formed by Leu8, Ile44, and Val70 of Ub$^{prox}$. Ser498 of yNpl4 forms a hydrogen bond with Arg42 of Ub$^{prox}$. The main-chain CO group of Gln493 in yNpl4 forms a hydrogen bond with the main-chain NH group of Leu71 in Ub$^{prox}$. The functional importance of these interactions was confirmed by SPR analyses using the A494F, S498L, and S498R mutants of His$_6$-yNpl4$^{113-580}$ (Table 2). The S498L mutation of yNpl4 had a little effect on the affinity for K48-, K63-, and M1-Ub$_4$, whereas the S498R mutation of yNpl4 decreased the affinity for K48-Ub$_4$ to 36% of the wild-type affinity (Table 2). These results suggest that the N loop of yNpl4 CTD is a binding site for Ub$^{prox}$, although the contribution of the N loop to K48-Ub$_4$ binding is smaller than that of the C-terminal helix. Reflecting this, gaps are found in the interface between yNpl4 and Ub$^{prox}$. One of the gaps is located between Ala494 of yNpl4 and Leu8 of Ub$^{prox}$. The Phe replacement of Ala494 in yNpl4 increased the affinity of yNpl4 for K48-Ub$_4$ 2.2-fold (Table 2). It is likely that the bulky side chain of Phe filled the gap and increased the affinity. The involvement of the N loop of yNpl4 CTD in the Ub$^{prox}$ recognition was further supported by GST pulldown assays using the Cdc48–UN complex containing a GST-yNpl4 mutant: the A494F mutation of yNpl4 enhanced the binding between Cdc48–UN and K48 chains, whereas the S498R mutation decreased it (Fig. 2d). Similar results were obtained even when the GST tag was fused to Ufd1 instead of Npl4 (Supplementary Fig. 3b).

The S498R or A494F mutation hardly affected the affinity for K63- or M1-Ub$_4$, in contrast to that for K48-Ub$_4$, and thereby

changed the linkage specificity (represented by the reciprocal ratio of dissociation constants; Table 2). The affinity of wild-type yNpl4 for K48-Ub$_4$ was 11 and 15 times higher than that for K63- or M1-Ub$_4$, respectively. The S498R mutation of the yNpl4 decreased the linkage specificity. In contrast, the A494F mutant of yNpl4 increased the linkage specificity because this mutant increases the affinity for K48-Ub$_4$ but hardly affects that for K63- or M1-Ub$_4$ (Table 2). These results indicate that the interaction between Ub$^{prox}$ and the N loop of yNpl4 CTD contributes to the specificity of yNpl4 to K48 chains.

**K48 chain–CTD binding stimulates the Cdc48 ATPase activity.** Lys-48-linked polyubiquitylated GFP (K48-Ub$_n$-GFP) has been used as a model substrate that can stimulate the ATP hydrolysis of the Cdc48–UN complex[13,14]. To evaluate the coupling between the binding to a K48 chain and ATP hydrolysis in the Cdc48–UN complex, we analyzed the rate of the ATP hydrolysis by the Cdc48–UN complex containing mutant yNpl4. Addition of K48-Ub$_n$-GFP increased the ATP hydrolysis rate of the Cdc48–UN complex, depending on the affinity of yNpl4 for K48 chains (Fig. 2e and Supplementary Fig. 3e). For instance, K48-Ub$_n$-GFP increased the ATP hydrolysis rate of the Cdc48–UN complex approximately 5-fold. The T571A, M574Q, or I575A mutation at the yNpl4–Ub$^{dist}$ interface, which severely decreases the affinity for K48 chains, decreased the ATP hydrolysis rate. The S498R mutation at the yNpl4–Ub$^{prox}$ interface, which shows a mild effect on the affinity, also decreased the ATPase hydrolysis rate but less than the T571A, M574Q, or I575A mutation. These results indicate that the stimulation of the ATPase activity of the Cdc48–UN complex requires the binding activity of yNpl4 CTD to K48 chains.

**Structure of Npl4 in complex with Ufd1.** Npl4 and Ufd1 can form a heterodimer, even in the absence of Cdc48/p97 or a polyubiquitylated substrate. The mechanism of the Ufd1–Npl4 interaction remains unclear, although residues 258–275 of human Ufd1 (hUfd1$^{258–275}$, with the prefix "h" indicating the human protein; equivalent to residues 288–305 of yUfd (yUfd1$^{288–305}$)) have been assigned as the Npl4-binding motif (NBM) (Fig. 1)[35]. Indeed, fluorescence anisotropy-based affinity measurement using FlAsH-labeled yUfd1$^{288–305}$ showed that yUfd1$^{288–305}$ binds to yNpl4 with $K_d$ of 85.7 nM (Table 3). To further reveal the structural basis of the interaction between Npl4 and Ufd1, we determined the crystal structure of yNpl4$^{113–580}$ (E123A K124A E125A) in complex with yUfd1$^{288–305}$ (Fig. 3a, b and Table 1). The structure was determined by the molecular replacement method using yNpl4$^{113–580}$ alone as the search model. In the crystal, yNpl4$^{113–580}$ formed a stoichiometric complex with yUfd1$^{288–305}$. No large conformational difference was observed between yNpl4$^{113–580}$ alone and the yUfd1$^{288–305}$-bound yNpl4$^{113–580}$ (Supplementary Fig. 2b).

In the complex, residues 298–300 of yUfd1 NBM form a β-sheet with Ins-1 of yNpl4 (Fig. 3c, d). Leu296, Phe326, Leu353, Met357, Phe419, Pro420, and Tyr424 of yNpl4 form a hydrophobic groove to accommodate Pro289, Leu292, Leu294, Gly297, Leu299, Phe301, Phe303, and Met305 of yUfd1. Mutations of all these residues except for Pro420 of yNpl4 and Pro289 of yUfd1 were examined by fluorescent anisotropy-based binding analysis (Table 3). The mutation of Leu296, Leu353, or Tyr424 of yNpl4 or Leu292, Gly297, Leu299, or Phe301 of yUfd1 severely affected the affinity between yUfd1$^{288–305}$ and yNpl4$^{113–580}$. The Ala replacement of Leu292, Leu299, or Phe301 in yUfd1$^{288–305}$ decreased the affinity to 1.6, 0.68, or 0.41% of the wild-type affinity, respectively. The Leu, Arg, or Tyr replacement of Gly297 of yUfd1$^{288–305}$ decreased the affinity for yNpl4$^{113–580}$ to 14, 0.41, or 4.0% of the wild-type affinity, respectively. The

| Table 3 Binding affinity of yNpl4 for yUfd1. | | |
|---|---|---|
| | $K_d$ (nM) | Fold of increase |
| WT #1[a] | 85.7 ± 2.6 | 1 |
| WT #2[a] | 99.4 ± 6.7 | 1 |
| yNpl4 WT + yUfd1 mutants | | |
| L292A | 5292 ± 788 | 62 |
| L294A | 527 ± 98 | 6 |
| E296A | 437 ± 48 | 5 |
| G297L | 619 ± 90 | 7 |
| G297R | 20896 ± 2066 | 244 |
| G297Y | 2136 ± 362 | 25 |
| L299A | 12677 ± 639 | 148 |
| F301A | 20770 ± 2906 | 242 |
| F303A | 1766 ± 116 | 21 |
| M305A | 261 ± 8 | 3 |
| yNpl4 mutants + yUfd1 WT | | |
| L296A | 7019 ± 60 | 82 |
| F326A | 257 ± 27 | 3 |
| L353A | 1380 ± 81 | 16 |
| M357A | 164 ± 14 | 2 |
| R364A | 314 ± 41 | 4 |
| F419A | 158 ± 18 | 2 |
| Y424A | 8447 ± 823 | 99 |

Data are presented as mean ± standard deviation; $n = 3$ independent experiments
[a]The assays of wild-type yNpl4 and yUfd1-FlAsH were performed twice using distinct samples with similar results

backbone dihedral angles φ and ψ of Gly297 are 90.1° and −8.5°, respectively. MolProbity[36], a standard program for protein structure validation, judges these angles as favored angles for Gly but allowed angles for non-Gly/Pro residues. Gly is more favorable than non-Gly/Pro residues at residue 297. yUfd1 NBM is kinked at Gly297, thereby fitting into the hydrophobic groove in the MPN subdomain of yNpl4 (Fig. 3b). Gly297 and Phe301 of yUfd1 are conserved from yeast to human, whereas Leu299 of yUfd1 is replaced by functionally equivalent hydrophobic residues among other eukaryotes (Fig. 3c). Although the sequence of Ufd1 NBM is variable among eukaryotes, this GxΦxF motif (x and Φ represent any amino acids and hydrophobic amino acids, respectively) is well conserved, reflecting its functional importance (Fig. 3c). The Ala replacement of Leu296, Phe326, Leu353, or Tyr424 of yNpl4$^{113–580}$ decreased the affinity to 1.2, 33, 6.2, or 1.0% of the wild-type affinity, respectively (Table 3). Notably, Leu296, Phe326, and Tyr424 of yNpl4 are conserved in hNpl4 (Supplementary Fig. 4). Leu296 and Phe326 of yNpl4 interact with Leu299 and Phe301 of yUfd1, which occupy the third and fifth positions of the GxΦxF motif, and Tyr424 of yNpl4 hydrophobically interacts with the main-chain CO group of Ala290 and the Cα atom of Lys291 in yUfd1. These hydrophobic interactions are likely conserved from yeast to human. In addition to the hydrophobic interactions, Arg364 and Thr418 of yNpl4 form hydrogen bonds with Glu296 and the main-chain NH group of Ala290 in yUfd1, respectively. The E296A mutation of yUfd1$^{288–305}$ or the R364A mutation of yNpl4$^{113–580}$ decreased the affinity to 20% or 27% of the wild-type affinity, respectively (Table 3). These residues are not conserved between yeast and human. The hydrophilic interactions between Npl4 and Ufd1 may be variable among species.

**Ufd1 does not overlap with K48 chain on Npl4.** The catalytic groove in the MPN domain of JAMM-family DUBs accommodates the C-terminal tail of Ub for cleavage[25,26,37]. The first insertion in the JAMM core (i.e., Ins-1) forms a β-sheet with the C-terminal tail of Ub in the catalytic groove (Fig. 3d). Npl4 is enzymatically inactive but has Ins-1 and a groove similar to the

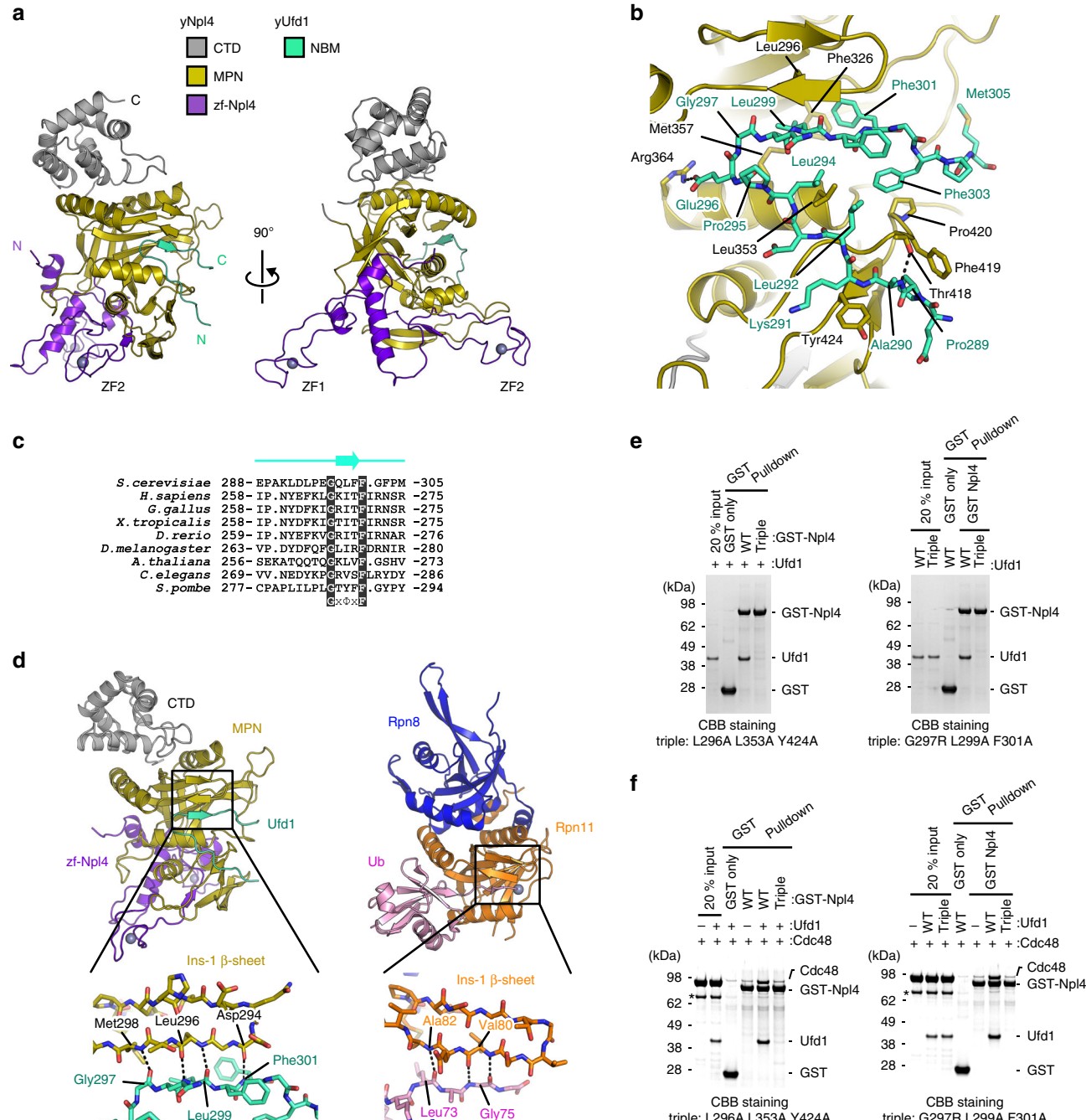

**Fig. 3 Crystal structure of yNpl4 in complex with yUfd1. a** Overall structure of the yNpl4–yUfd1 complex in two orientations. **b** Close-up view of the interaction between yNpl4 and yUfd1. Hydrogen bonds are shown as black dotted lines. **c** Sequence alignment of the NBM region of Ufd1. The position of the β-strand in the NBM region in the yNpl4–yUfd1 structure is shown above the sequences. **d** Comparison of the MPN region of the yNpl4–yUfd1 and Rpn8–Rpn11–Ub (PDB 5U4P [https://doi.org/10.2210/pdb5u4p/pdb])[25] complexes. The coloring scheme of the yNpl4–yUfd1 complex is the same as that in Fig. 3a. Rpn8, Rpn11, and Ub of the Rpn8–Rpn11–Ub complex are blue, orange, and pink, respectively. Hydrogen bonds are shown as black dotted lines. **e, f** Analysis of the binding between GST-yNpl4 and yUfd1 alone (**e**) or yUfd1–Cdc48 (**f**) by pulldown assays. The results from the triple mutant of yNpl4 (L296A L353A Y424) (left) or the triple mutant of yUfd1 (G297R L299A F301A) (right) are shown. In all, 20% input means 20% of the volume of the sample (yUfd1 and/or Cdc48) that was mixed with the GST-yNpl4-bound glutathione resin. Asterisks indicate contamination. The bound proteins were analyzed by SDS-PAGE and stained with Coomassie brilliant blue. Source data are provided as a Source Data file.

catalytic groove of JAMM-family DUBs[19]. Therefore, this groove and Ins-1 have been assumed to form a potential Ub chain-binding site. However, the present yNpl4$^{113–580}$–yUfd1$^{288–305}$ structure reveals that the groove of the MPN domain accommodates yUfd1 NBM (Fig. 3d). The Npl4-bound yUfd1 does not

overlap with K48-Ub$_2$ in the yNpl4–K48-Ub$_2$ complex. Consistently, yUfd1 did not inhibit the binding of K48 chains to yNpl4 (Fig. 2d, Supplementary Fig. 3d). In addition, the recently reported cryo-EM structure of the substrate-engaged Cdc48-UN showed that the bound K48 chain and Ufd1 do not overlap with

each other. In spite of the similarity of the MPN domains of Npl4 and JAMM DUBs, the functions of Ins-1 and the groove are different between them.

**Cdc48–UN assembly depends on the Ufd1–Npl4 interaction.** To analyze the relationship between the Ufd1–Npl4 interaction and Cdc48–UN assembly, we first screened for mutations that effectively inhibit the interaction between yUfd1 and yNpl4 by GST pulldown experiments with full-length proteins. Although single point mutations of yNpl4 or yUfd1 hardly affected the yNpl4–yUfd1 interaction (Supplementary Fig. 5a, b), triple mutations of yNpl4 (L296A L353A Y424A) or yUfd1 (G297R L299A F301A) completely inhibited the formation of the UN heterodimer, respectively (Fig. 3e and Supplementary Fig. 5a, b). The triple mutant yNpl4 was confirmed to bind to K48 chains as well as wild-type yNpl4 (Supplementary Fig. 5c). Next, we analyzed the effect of yNpl4 or yUfd1 mutations on the Cdc48–UN assembly by GST pulldown experiments and found that the triple mutants of either yNpl4 or yUfd1 substantially reduced the formation of the Cdc48–UN complex (Fig. 3f and Supplementary Fig. 5d). Thus, the UN heterodimer formation facilitates the Cdc48–UN assembly, although both yNpl4 and yUfd1 can directly bind to Cdc48.

To further assess the functional significance of the Ufd1–Npl4 interaction in vivo, mutated Npl4-3xFLAG or Ufd1-3xFLAG was expressed in npl4Δ or ufd1Δ cells, respectively. We examined the accumulation of Ub conjugates in total cell lysate in these cells. The npl4-1 temperature-sensitive strain was also examined as an Npl4-deficient control. The levels of Ub conjugates were not changed in the cells that expressed yNpl4 or yUfd1 single-point mutants (Supplementary Fig. 5e). In contrast, in the cells expressing the triple mutant of yNpl4 or yUfd1, we observed a substantial accumulation of Ub conjugates, which was comparable to that in the npl4-1 cell (Supplementary Fig. 5e). Correspondingly, these triple mutant-expressing cells displayed a temperature-sensitive growth phenotype, although it was milder than the npl4-1 phenotype (Supplementary Fig. 5f). These findings indicate that the Ufd1–Npl4 interaction is important for the degradation of Ub conjugates and for the cell growth.

**CTD of human Npl4 is involved in binding to K48 chains.** Npl4 is a highly conserved protein from yeast to mammals. In mammalian Npl4, the NZF domain located in its C-terminus binds to K48 chains and Lys63-linked Ub chains (K63 chains). The human Npl4 (hNpl4) mutant lacking the NZF domain hardly binds to Ub chains[15]. On the other hand, the *Ascomycota* and *Plantae* Npl4 proteins lack the NZF domain in their C-termini (Supplementary Fig. 6a, b). The Npl4 proteins from *S. cerevisiae* and *C. thermophilum* bind to K48 chains via the zf-Npl4-MPN-CTD domain[19]. Multiple amino-acid sequence alignment including yNpl4 and hNpl4 shows that the Ub$^{dist}$-interacting residues in the C-terminal helix of the CTD are partially conserved between yeast and human; Thr571 of yNpl4 is conserved, and Ile575 of yNpl4 is replaced with Leu in hNpl4, which should be functionally equivalent to Ile (Supplementary Fig. 6a). Intriguingly, the T551A mutation of hNpl4 (equivalent to Thr571 of yNpl4) decreased the affinity for K48-Ub$_4$ to 58% of the wild-type affinity (Table 2). On the other hand, Met574 of yNpl4 is replaced with Gln in hNpl4 and with Ala in *S. pombe*. The M574Q mutation of yNpl4 decreased the affinity for K48-Ub$_4$ to 9.1% of the wild-type affinity, whereas the M574A mutation decreased it to just 43% (Table 2). Reciprocally, the Q554M mutation of hNpl4 (equivalent to Met574 of yNpl4) increased the affinity for K48-Ub$_4$ about two-fold (Table 2). These findings suggest that the C-terminal

helix of hNpl4 CTD is also involved in binding to a Ub chain like that of yNpl4 CTD.

**hNpl4 NZF can compensate for the defect of yNpl4 CTD.** To assess the functional equivalence between yNpl4 CTD and hNpl4 NZF in the context of the Ub chain binding and the Cdc48 ATPase activity, we investigated engineered yNpl4 proteins, where the NZF domain of hNpl4 (including the linker region between the CTD and NZF domains) was fused to the C-terminal end of the full-length yNpl4 (yNpl4-NZF) with or without mutations deficient in binding to Ub chains (T571A or I575A). GST pulldown analysis showed that yNpl4-NZF bound to both K48 and K63 chains, even with the T571A or I575A mutation, indicating that the NZF fusion can rescue the defect of Ub chain binding of the CTD mutations (Fig. 4a, b and Supplementary Fig. 6c). Next, we analyzed whether the yNpl4-NZF proteins support the K48 chain-dependent ATPase activity of the Cdc48 complex (Fig. 4c and Supplementary Fig. 6d). The NZF fusion to wild-type yNpl4 did not enhance the ATPase activity above the wild-type level, probably because the affinity of wild-type yNpl4 for K48 chains is sufficient to stimulate the full ATPase activity of Cdc48. On the other hand, the NZF fusion completely recovered the decrease in the ATPase stimulation by the T571A or I575A mutation, suggesting that yNpl4 CTD and hNpl4 NZF are functionally equivalent. This also raises the possibility that the enhancement of ATP hydrolysis is independent of Lys48-linkage specificity. However, K63 chains did not stimulate the ATPase activity of the Cdc48–UN complex containing yNpl4-NZF, which can bind to K63 chains as well as to K48 chains (Fig. 4c). Thus, the stimulation of the ATPase activity of the Cdc48–UN complex actually depends on Lys48-linkage specificity of yNpl4. Considering that Lys48-linkage specificity of yNpl4 CTD is dispensable for the stimulation of the ATP hydrolysis, one may expect that yNpl4, yUfd1, and/or Cdc48 have additional Lys48-linkage-specific recognition site(s) besides Npl4 CTD. The cryo-EM structure of the substrate-engaged Cdc48–UN has shown that yNpl4 MPN binds to unfolded Ub, which bridges between Ub$^{prox}$ and the central pore of Cdc48[22] (Supplementary Fig. 6e). This binding may be related to the K48 chain-specific stimulation of the ATPase activity of Cdc48.

**Discussion**
In this study, we determined the crystal structure of the yNpl4$^{113–580}$–K48-Ub$_2$ complex. Both Ub$^{dist}$ and Ub$^{prox}$ interact with yNpl4 CTD, which is a newly identified Ub-binding domain (Fig. 5a). The C-terminal helix of yNpl4 CTD is the primary Ub-binding site of yNpl4. Among the previously reported Ub-binding domains (UBDs), Ub-interacting motif (UIM), motif interacting with Ub (MIU), and UIM and MIU related UBD (UMI) have a single helix as a Ub-binding site. yNpl4 CTD, MIU, and UMI but not UIM exhibit the same helix orientation relative to the bound Ub. Therefore, we compared the Ub-binding helices of yNpl4 CTD, RNF168 UMI (PDB 5XIS [https://doi.org/10.2210/pdb5xis/pdb])[38], and Rabex5 MIU (PDB 2C7N [https://doi.org/10.2210/pdb2c7n/pdb])[39] (Fig. 5b). These UBDs have an N-terminal hydrophilic residue for hydrogen bonding with the NH groups of Ala46 and Gly47 of Ub, and a central hydrophobic residue for the interaction with the hydrophobic pocket formed by Leu8, Ile44 and Val70 of Ub (Fig. 5b, c). The differences of the N-terminal hydrophilic residue and central hydrophobic residue in the Ub-binding helix affect its orientation relative to Ub. The central hydrophobic residues are Ile in Npl4 CTD and UMI and Ala in MIU. The longer side chain of Ile in yNpl4 CTD or UMI pushes out the C-terminal end of the helical UBD, whereas those N-terminal hydrophilic residues move close proximity to the NH groups of Ala46 and Gly47 of Ub.

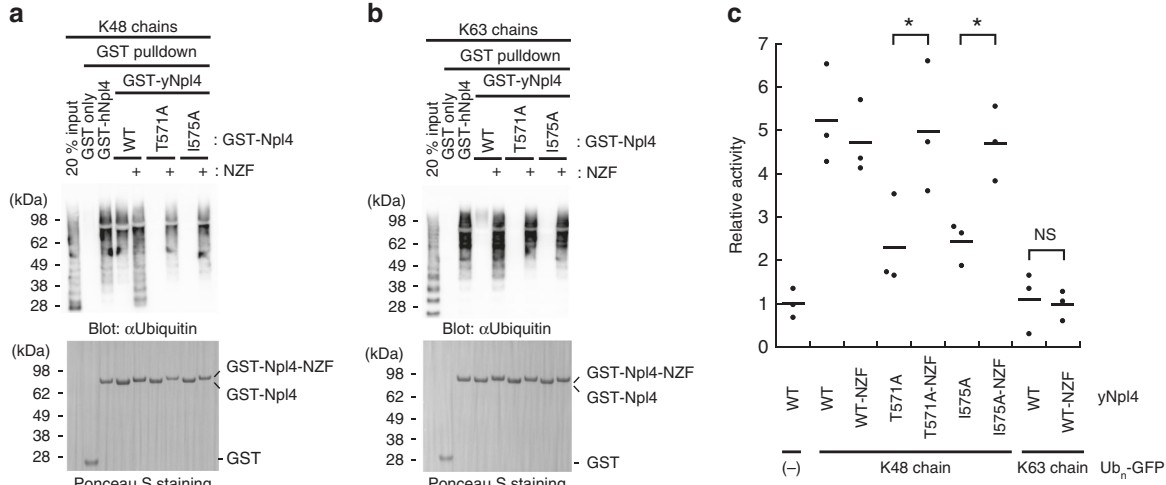

**Fig. 4 Analysis of the NZF-fused yNpl4. a**, **b** Analysis of the binding of GST-hNpl4, GST-yNpl4 or GST-yNpl4-NZF to K48 chains (**a**) or K63 chains (**b**) by pulldown assays. The bound Ub chains were detected by immunoblotting with anti-Ub antibody (upper panel). Blot membranes were stained with Ponceau S (lower panel). 20% input means that 20% of the volume of the sample (Ub chains) that was mixed with GST-yNpl4- or GST-yNpl4-NZF-bound glutathione resin. These experiments were repeated with distinct samples (Supplementary Fig. 6c). Source data are provided as a Source Data file. **c** ATP hydrolysis rates of the Cdc48–UN complex containing yNpl4 or yNpl4-NZF with K48-$Ub_n$-GFP or K63-$Ub_n$-GFP. The rates were normalized to the average of the ATP hydrolysis rates of the wild-type Cdc48–UN complex without $Ub_n$-GFP. The line represents the mean of the rates after normalization (mean values; $n = 3$ independent experiments; *$P < 0.05$ from Student's $t$-test).

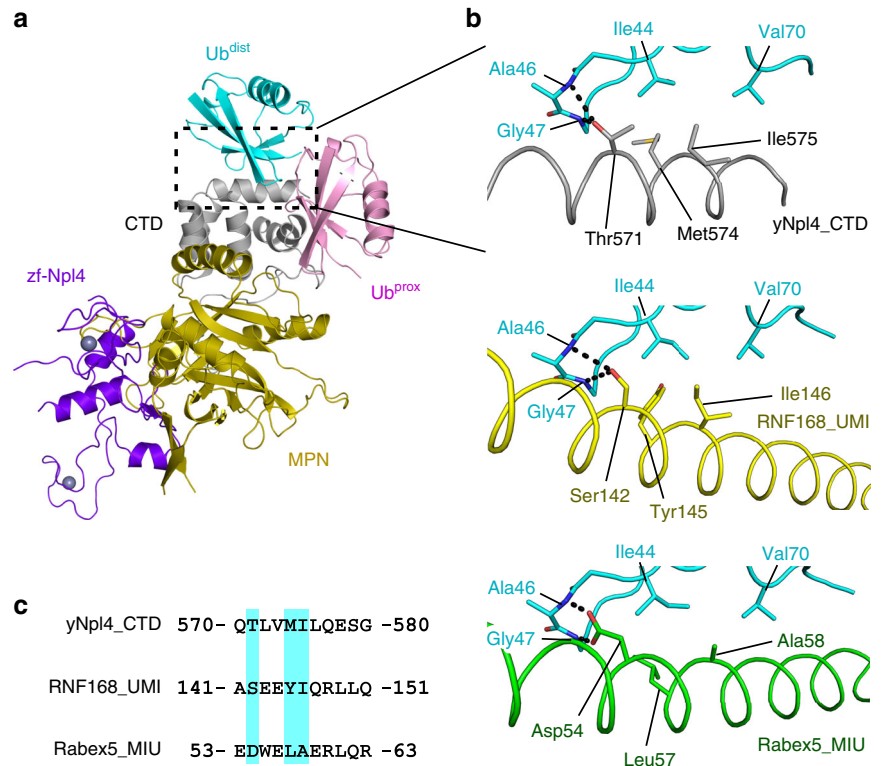

**Fig. 5 Comparison of the C-terminal helix of yNpl4, RNF168 UMI, and Rabex-5 MIU. a** Overall structure of the yNpl4–K48-$Ub_2$ complex. The coloring scheme is the same as that in Fig. 2a. **b** Structural comparison of the Ub-binding interfaces of the C-terminal helix of yNpl4 (this study), RNF168 UMI (PDB 5XIS [https://doi.org/10.2210/pdb5xis/pdb])[38], and Rabex-5 MIU (PDB 2C7M [https://doi.org/10.2210/pdb2c7m/pdb])[39]. The coloring scheme of the yNpl4–Ub$^{dist}$ complex is the same as that in Fig. 2a. RNF168 UMI and Rabex-5 MIU are colored yellow and green, respectively. Hydrogen bonds are shown as black dotted lines. The residues involved in the binding between yNpl4 and Ub$^{dist}$ and the corresponding residues of RNF168 UMI and Rabex-5 MIU are shown as sticks. **c** Sequence alignment of the C-terminal helix of yNpl4, RNF168 UMI, and Rabex-5 MIU. The residues involved in the binding between yNpl4 and Ub$^{dist}$ and the corresponding residues of RNF168 UMI and Rabex-5 MIU are highlighted in cyan.

Because of this proximity, the N-terminal hydrophilic residue of yNpl4 CTD and UMI are Thr and Ser, respectively, which are shorter than Asp, the N-terminal hydrophilic residue of MIU.

The cryo-EM structure of the substrate-engaged Cdc48–UN[22] contains two folded Ub moieties, which correspond to $Ub^{dist}$ and $Ub^{prox}$ in the present crystal structure of $yNpl4^{113-580}$–$K48$-$Ub_2$ (Fig. 6a). In both structures, the C-terminal helix of yNpl4 CTD interacts with the Ile44-centered hydrophobic patch of $Ub^{dist}$ (Fig. 6b), and therefore, the $Ub^{dist}$ recognition mode of yNpl4 is essentially the same. On the other hand, the $Ub^{prox}$ recognition mode is different. In the yNpl4–$K48$-$Ub_2$ structure, Ser498 of yNpl4 forms a hydrogen bond with Arg42 of $Ub^{prox}$. GST pull-down and SPR analysis of the S498R mutant of yNpl4 showed that Ser498 of yNpl4 is actually involved in binding to K48 chains (Fig. 2d, Supplementary Fig. 3b and Table 2). In the structure of the substrate-engaged Cdc48–UN, $Ub^{prox}$ moves toward Cdc48 by ~10 Å, as compared with yNpl4–$K48$-$Ub_2$, and does not interact with Ser498 of yNpl4 (Fig. 6b). This $Ub^{prox}$ movement appears to be coupled with threading of the unfolded Ub moiety; i.e., $Ub^{prox}$ interacts with Ser498 of yNpl4 at the initial Ub chain recognition step. At the subsequent substrate-threading step, the unfolded Ub is inserted into the central pore of Cdc48, and pulls $Ub^{prox}$, which loses its interaction with Ser498 of yNpl4.

We also determined the crystal structure of $yNpl4^{113-580}$–$yUfd1^{288-305}$. We docked the structure of the NBM region of Ufd1 into the cryo-EM structure of the substrate-engaged Cdc48–UN[22] (Fig. 6c). The present $yUfd1^{288-305}$ model was nicely fitted into the unassigned density observed in the cryo-EM map (Fig. 6c, d), which was supposed to correspond to a part of yUfd1[22]. This indicates that the yNpl4–Ufd1 interaction revealed by crystallography of the UN complex similarly occurs in the substrate-engaged Cdc48–UN (Fig. 6d).

Inhibitors of Ub-proteasome system (UPS) have recently been developed as anticancer drugs[40]. Because the p97/Cdc48–UN complex plays crucial roles upstream of the proteasome, its inhibition has emerged as a novel therapeutic target in cancer cells[41]. However, since p97/Cdc48 also plays various roles outside the UPS pathway, its inhibition may affect many cell functions. In the context, disruptions of the specific p97-cofactor association will likely lead to the novel p97 inhibitors with enhanced specificity of anticancer activity. Inhibition of the interaction between Ufd1 and Npl4 prevents formation of the Cdc48-UN complex (Fig. 3f). Our present structure of $yNpl4^{113-580}$–$yUfd1^{288-305}$ might facilitate the development of such anticancer drugs by serving as a useful platform for structure-based design.

## Methods

**Preparation of Npl4, Ufd1, and Cdc48.** The codon-optimized cDNAs of yNpl4, hNpl4, and yUfd1 were synthesized (Eurofins) to improve their expressions in *Escherichia coli*. For crystallization, the gene encoding $yNpl4^{113-580}$ or $yUfd1^{288-305}$ was cloned into the pGEX-6P1 expression vector using *Bam*HI and *Xho*I sites or pCold-GST expression vector using *Nde*I and *Xho*I sites, respectively, to produce the N-terminal GST fusion proteins. For GST pulldown assays, the full-length yNpl4 and yUfd1 genes were cloned into the pGEX-6P1 expression vector using *Bam*HI and *Xho*I sites[15], and the Cdc48 gene was cloned into the pET21a expression vector using *Bam*HI and *Xho*I sites. For SPR analyses, the gene encoding $yNpl4^{113-580}$ or $hNpl4^{105-608}$ was cloned into the pET28a expression vector using *Nde*I and *Xho*I sites to produce the N-terminal $His_6$-tagged proteins. For fluorescence anisotropy-based affinity measurements, the GWCCPGCC sequence was attached to the C-terminus of $yUfd1^{288-305}$. The $yUfd1^{288-305}$-GWCCPGCC gene was cloned into the pCold-SUMO expression vector using *Nde*I and *Xho*I sites to produce the N-terminal $His_6$-tagged SUMO fusion protein. Mutations were generated by PCR-based mutagenesis. Primer sequences used in this study are shown in Supplementary Table 1. *E. coli* strain Rosetta (DE3) cells (Invitrogen) were transformed with each expression vector, and cultured in LB medium containing 100 mg $L^{-1}$ ampicillin for the pGEX-6P1 or pCold expression vectors or 50 mg $L^{-1}$ kanamycin for the pET28a expression vector at 37 °C. When the optical density of the culture at 600 nm reached ~0.5, isopropyl-β-D-thiogalactopyranoside (IPTG) was added to a final concentration of 0.1 mM to induce protein expression, and the culture was further continued for 18 h at 20 °C for the

pGEX-6P1 and pET28a expression vectors and for 24 h at 15 °C for the pCold expression vector. The cells transformed with the pGEX-6P1 or pCold-GST expression vector were disrupted by sonication in phosphate buffered saline (PBS) containing 1 mM dithiothreitol (DTT) and 0.5% Triton X-100, and purified by a Glutathione Sepharose FF column (GE Healthcare) and a Resource Q anion exchange column (GE Healthcare). The GST tag of GST-$yNpl4^{113-580}$ was cleaved by HRV3C protease, and the sample was further purified by a Resource Q anion exchange column and a HiLoad 16/60 Superdex 75 column (GE Healthcare) in 10 mM Tris-HCl buffer (pH 7.2) containing 50 mM NaCl and 5 mM β-mercaptoethanol. To prepare the $yNpl4^{113-580}$–$yUfd1^{288-305}$ complex for crystallization, a two-fold molar excess of GST-$yUfd1^{288-305}$ was incubated at 4 °C for 10 min with $yNpl4^{113-580}$. The GST-$yUfd1^{288-305}$–$yNpl4^{113-580}$ complex was purified by a HiLoad 16/60 Superdex 75 column in 10 mM Tris-HCl buffer (pH 7.2) containing 50 mM NaCl and 5 mM β-mercaptoethanol. The GST tag of GST-$yUfd1^{288-305}$–$yNpl4^{113-580}$ was cleaved by HRV3C protease. To remove GST, the sample was passed over a Glutathione Sepharose FF column pre-equilibrated with 10 mM Tris-HCl buffer (pH 7.2) containing 50 mM NaCl and 5 mM β-mercaptoethanol. The cells transformed with the pCold-SUMO or pET28a expression vectors were disrupted by sonication in 50 mM Tris-HCl buffer (pH 8.0) containing 150 mM NaCl and 0.5% Triton X-100, and purified by a nickel-nitrilotriacetic acid (Ni-NTA) column (Qiagen) and a Resource Q anion exchange column (GE Healthcare), except for Cdc48. For preparation of Cdc48, the Cdc48-expressing cells were lysed in lysis buffer (50 mM sodium phosphate buffer (pH 7.0) containing 300 mM NaCl, 10% glycerol, 1 mM Tris (2-carboxiehyl) phosphine hydrochloride (TCEP), 5 mM $MgCl_2$, and 100 μM ATP) and disrupted by sonication. After addition of Triton X-100 (final concentration, 0.1%), the lysate was clarified by centrifugation at 29,300 x g for 30 min. The resultant supernatant was incubated with TALON resin (Takara). After extensive washing, Cdc48 was eluted in 50 mM HEPES-NaOH buffer (pH 7.1) containing 100 mM NaCl, 0.3 M imidazole, 5 mM $MgCl_2$, 100 μM ATP, and 0.5 mM TCEP. To further enrich for hexameric Cdc48, the solution was loaded on a Superose 6 10/300 column equilibrated to 50 mM HEPES-NaOH buffer (pH 7.5) containing 100 mM NaCl, 5 mM $MgCl_2$, and 0.5 mM TCEP with a flow rate of 0.25 mL $min^{-1}$. To prepare the $yUfd1$-$His_6$-$yNpl4^{113-580}$ complex for SPR analysis, the cells expressing GST-yUfd1 and those expressing $His_6$-$yNpl4^{113-580}$ were mixed and disrupted at the same time by sonication in PBS containing 1 mM DTT and 0.5% Triton X-100. The cleared lysate was loaded onto a Glutathione Sepharose FF column (GE Healthcare). The GST tag of GST-yUfd1–$His_6$-$yNpl4^{113-580}$ was cleaved by HRV3C protease, and the sample was further purified by a Ni-NTA column (Qiagen) and a HiLoad 16/60 Superdex 75 column (GE Healthcare) in 10 mM HEPES-NaOH (pH 7.5) containing 150 mM NaCl.

**Preparation of Ub chains and ubiquitylated substrates.** Ub was overproduced in *E. coli* strain Rosetta (DE3) cells (Invitrogen) transformed with the pET26b expression vector harboring the Ub gene in LB medium containing 50 mg $L^{-1}$ kanamycin at 20 °C. The SeMet-labeled Ub, Ub (P19M V26M), and Ub (I30M) was overproduced in the methionine-auxotroph *E. coli* strain B834 (DE3) cells in the customized medium equivalent to LeMaster medium (Code No. 06780, Nacalai tesque) with 30 μg $mL^{-1}$ L-SeMet (Nacalai tesque) and 50 mg $L^{-1}$ kanamycin at 20 °C. The cells were disrupted by sonication in 50 mM ammonium acetate buffer (pH 4.5). The cleared lysates of Ub and Ub variant were incubated for 5 min at 80 and 60 °C, respectively. The denatured and insolubilized *E. coli* proteins were pelleted by centrifugation at 30,000×g for 60 min. The supernatant was purified by a Resource S cation exchange column (GE Healthcare) and a HiLoad 26/60 Superdex 75 size-exclusion column (GE Healthcare) in 10 mM Tris-HCl buffer (pH 7.2) containing 50 mM NaCl. The purified Ub was concentrated with an Amicon Ultra-15 10,000 MWCO filter (Millipore).

$K48$-, $K63$-, and $M1$-$Ub_4$, and SeMet-labeled $K48$-$Ub_2$ were synthesized enzymatically. For $K48$-$Ub_4$ synthesis, E1 (0.25 μM), E2-25K (5 μM), and Ub (2 mM) were mixed in the reaction buffer (50 mM Tris-HCl (pH 9.0) containing 10 mM ATP, 10 mM $MgCl_2$, and 0.6 mM DTT) and incubated at 37 °C for 15 h. For SeMet-labeled $K48$-$Ub_2$ synthesis, E1 (0.25 μM), E2-25K (5 μM), and Ub (2 mM) were mixed in the reaction buffer and incubated at 37 °C for 15 h. For $K63$-$Ub_4$ synthesis, E1 (0.25 μM), Ubc13 (10 μM), MMS2 (10 μM) and Ub (2 mM) were mixed in the reaction buffer and incubated at 37 °C for 15 h. For $M1$-$Ub_4$ synthesis, E1 (0.2 μM), UbcH7 (5 μM), HOIP (residues 697–1072; 0.1 μM) and Ub (2 mM) were mixed in the reaction buffer and incubated at 37 °C for 15 h. Each reaction solution was mixed with four volumes of 50 mM ammonium acetate buffer (pH 4.5) and loaded onto a Resource S cation exchange column (GE Healthcare) pre-equilibrated with 50 mM ammonium acetate buffer (pH 4.5) containing 140 mM NaCl. The synthesized $Ub_4$ species or SeMet-labeled $K48$-$Ub_2$ were eluted with a linear gradient of 140-400 mM NaCl in 50 mM ammonium acetate buffer (pH 4.5). Peak fractions containing the $Ub_4$ species or SeMet-labeled $K48$-$Ub_2$ were loaded onto a HiLoad 16/60 Superdex 75 size-exclusion column (GE Healthcare) with 10 mM HEPES-NaOH (pH 7.5) containing 150 mM NaCl. The purified Ub chains were concentrated to ~1 mM and stored at −80 °C until use.

For preparation of $Ub_n$-GFP substrates, the gene encoding Ub-sfGFP-cytochrome*b*2 derived tail (Ub-GFP)[42] was cloned into the pET21a expression vector using *Nde*I and *Eco*RI sites. $Ub_n$-GFP substrates were enzymatically synthesized and purified by column chromatography[14]. For $K48$-$Ub_n$-GFP

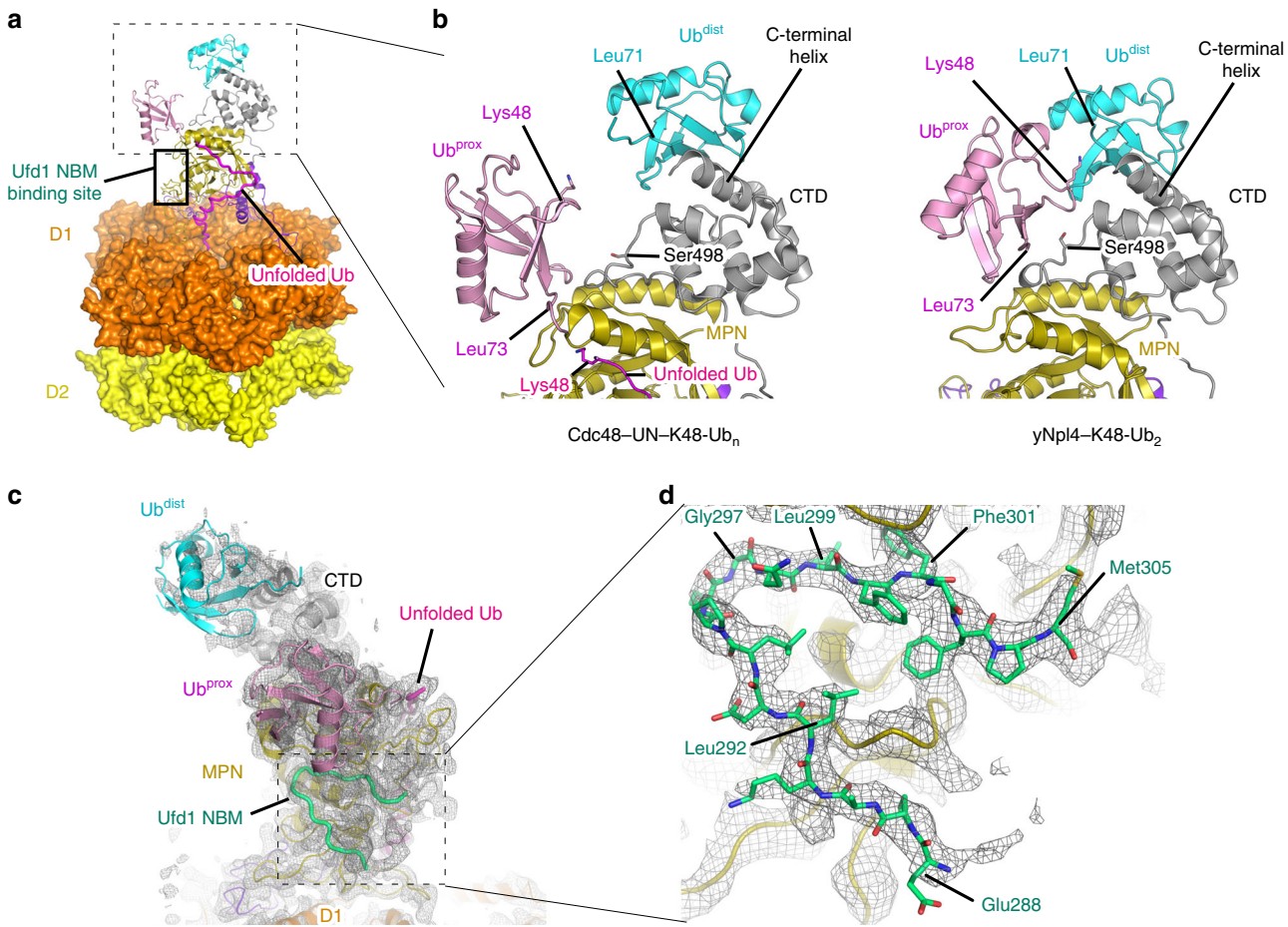

**Fig. 6 Comparison of the structures of yNpl4–K48-Ub₂, yNpl4–yUdf1 and the substrate-engaged Cdc48-UN complex. a** Overall view of the cryo-EM structure of the Cdc48-UN complex (PDB 6OA9 [https://doi.org/10.2210/pdb6oa9/pdb])[22]. The coloring scheme is the same as that in Fig. 1. The unfolded Ub is colored magenta. **b** Comparison of the K48 chain-binding site of yNpl4–K48-Ub₂ and the substrate-engaged Cdc48-UN complex. The coloring scheme is the same as that in Fig. 6a. **c**, **d** The structure of yNpl4–yUfd1 was superposed on the substrate-engaged Cdc48-UN complex using the Coot LSQ protocol with the zf-Npl4-MPN-CTD structure of yNpl4 as the reference[47]. yUfd1 NBM was further fitted as a rigid body into the cryo-EM map of the substrate-engaged Cdc48-UN (EMD-0665)[22] and was refined against it with real space refinement in Phenix[45]. The coloring scheme is the same as that in Fig. 3a. The cryo-EM map is shown as grey mesh. **c** Overall view of yNpl4–yUfd1 and two folded Ub moieties with the cryo-EM map contoured at 5 σ level. **d** Close-up view of the interaction between yNpl4 and yUfd1 with the cryo-EM map contoured at 8 σ level.

synthesis, 10 μM Ub-GFP, 1 μM E1, 20 μM gp78RING-Ube2g2, and 400 μM Ub were incubated in 20 mM HEPES-NaOH buffer (pH 7.4) containing 10 mM ATP and 10 mM MgCl₂ at 37 °C overnight. For K63-Ubₙ-GFP synthesis, 20 μM Ubc13/MMS2 was used, instead of 20 μM gp78RING-Ube2g2. Ubₙ-GFP substrates were bound to Ni-NTA beads, washed, and then eluted in 50 mM HEPES-NaOH buffer (pH 7.5) containing 100 mM NaCl and 0.3 M imidazole. The elution was loaded onto a Superdex 200 increase 10/300 column equilibrated to 50 mM HEPES-NaOH (pH 7.5) buffer containing 100 mM NaCl and 10% glycerol. Fractions containing long Ub chains (approximately Ub₁₀ in average) were collected for ATPase assay.

**Crystallization**. For crystallization of yNpl4$^{113-580}$ and the yNpl4$^{113-580}$–yUfd1$^{288-305}$ complex, the E123A, K124A, and E125A mutations were introduced into yNpl4$^{113-580}$ so as to reduce excess surface conformational entropy[27]. For crystallization of the yNpl4$^{113-580}$–SeMet-labeled K48-Ub₂ (WT, P19M V26M, or I30M) complex, yNpl4$^{113-580}$ was mixed with the SeMet-labeled K48-Ub₂ in a molar ratio of 1:1.2. Initial crystallization screening was performed with the sitting drop vapor diffusion method at 20 °C using a Mosquito liquid-handling robot (TTP Lab Tech). We tested about 500 conditions with crystallization reagent kits supplied by Hampton Research and Qiagen. Initial hits were further optimized. The best crystals of yNpl4$^{113-580}$ were grown at 20 °C with the sitting drop vapor diffusion method by mixing 0.5 μL of protein solution with an equal amount of reservoir solution containing 4% Tacsimate (pH 6.0) and 12% PEG3350 and equilibration against 500 μL of the reservoir solution. The best crystals of the yNpl4$^{113-580}$–yUfd1$^{288-305}$ complex were grown at 20 °C with the sitting drop vapor diffusion method by mixing 0.2 μL of protein solution with an equal amount of reservoir solution containing 90 mM Bis-Tris-HCl buffer (pH 7.5), 19% PEG3350, and 10 mM ATP and equilibration against 50 μL of the reservoir

solution. The best crystals of the yNpl4$^{113-580}$–SeMet-labeled K48-Ub₂ (WT) complex were grown at 20 °C with the sitting drop vapor diffusion method by mixing 0.5 μL of protein solution with an equal amount of 100 mM Bicine-NaOH buffer (pH 9.0) containing 34% PEG3350 and 200 mM Li₂SO₄ and equilibration against 500 μL of reservoir solution containing 100 mM Bicine-NaOH buffer (pH 9.0), 18% PEG3350, and 200 mM Li₂SO₄. The best crystals of the yNpl4$^{113-580}$–SeMet-labeled K48-Ub₂ (P19M V26M) complex were grown at 20 °C with the sitting drop vapor diffusion method by mixing 0.5 μL of protein solution with an equal amount of 100 mM Bicine-NaOH buffer (pH 9.0) containing 24% PEG3350, 200 mM NaCl, 3% 1,5-diaminopentane, and 10 mM MgCl₂ and equilibration against 500 μL of reservoir solution containing 100 mM Bicine-NaOH buffer (pH 9.0), 20% PEG3350, and 200 mM NaCl. The best crystals of the yNpl4$^{113-580}$–SeMet-labeled K48-Ub₂ (I30M) complex were grown at 20 °C with the sitting drop vapor diffusion method by mixing 0.5 μL of protein solution with an equal amount of 100 mM Bicine-NaOH buffer (pH 9.0) containing 23% PEG3350, 200 mM Li₂SO₄, 3% 1,5-diaminopentane, and 10 mM MgCl₂ and equilibration against 500 μL of reservoir solution containing 100 mM Bicine-NaOH buffer (pH 9.0), 12% PEG3350, 200 mM Li₂SO₄, and 10 mM MgCl₂. For data collection, the crystals were transferred to cryostabilizing solution, which was the individual reservoir solution containing 30% glycerol for yNpl4$^{113-580}$, 20% glycerol for yNpl4$^{113-580}$–yUfd1$^{288-305}$, or saturated trehalose for yNpl4$^{113-580}$–SeMet-labeled K48-Ub₂ (WT, P19M V26M, or I30M). The cryoprotected crystals were flash frozen in liquid N₂.

**Structure determination**. Diffraction data sets were collected at beamline BL41XU in SPring-8 (Hyogo, Japan) at 100 K. PILATUS3 6 M (Dectris) was used for the data collection of yNpl4$^{113-580}$ and yNpl4$^{113-580}$–SeMet-labeled K48-Ub₂ and

EIGER X 16 M (Dectris) was used for the data collection of yNpl4$^{113-580}$–yUfd1$^{288-305}$. The wavelengths for the data collection were 1.28260, 0.97904, and 1.00000 Å for yNpl4$^{113-580}$, yNpl4$^{113-580}$–SeMet-labeled K48-Ub$_2$, and yNpl4$^{113-580}$–yUfd1$^{288-305}$, respectively. The data sets were processed with HKL2000[43] and the CCP4 program suite[44]. To solve the structure of yNpl4$^{113-580}$ from the SAD data set using zinc anomalous scattering, the program Phenix was used for heavy-atom search, phase calculation, and density modification[45]. The structures of the yNpl4$^{113-580}$–yUfd1$^{288-305}$ and yNpl4$^{113-580}$–SeMet-labeled K48-Ub$_2$ complexes were determined by the molecular replacement method using the program MolRep[46]. The crystal structure of yNpl4$^{113-580}$ was used as the search model. The solution of Ub moieties in the yNpl4–K48-Ub$_2$ structure was not found using the program MolRep with the crystal structure of Ub (PDB 1UBQ [https://doi.org/10.2210/pdb1ubq/pdb])[28] as the search model. We manually assigned the Ub models in residual density using the program Coot[47]. The position of K48-Ub$_2$ was confirmed by the anomalous difference Fourier map of selenium atoms in the yNpl4–K48-Ub$_2$ (P19M V26M) and yNpl4–K48-Ub$_2$ (I30M) structures (Supplementary Fig. 1c). The atomic models were corrected using Coot[47] with careful inspection. Refinement was carried out using Phenix with iterative correction and refinement of the atomic models. There is one copy of the structure in the asymmetric unit of the yNpl4$^{113-580}$ or yNpl4$^{113-580}$–yUfd1$^{288-305}$ crystal. On the other hand, there are two yNpl4$^{113-580}$ and two Ub (Ub$^{dist}$ and Ub$^{prox}$) molecules in the asymmetric unit of the yNpl4$^{113-580}$–SeMet-labeled K48-Ub$_2$ crystal. Torsion-angle NCS restrains were applied during refinement of the yNpl4–SeMet-labeled K48-Ub$_2$ structure. The final models of yNpl4$^{113-580}$, yNpl4$^{113-580}$–yUfd1$^{288-305}$, and yNpl4$^{113-580}$–SeMet-labeled K48-Ub$_2$ have excellent stereochemistry (Table 1). All molecular graphics were prepared with PyMOL (DeLano Scientific; http://www.pymol.org).

**Sequence alignment.** Multiple sequence alignment was performed using the program ClustalW[48]. The figure of the sequence alignment between yNpl4 and hNpl4 was prepared using the program ESPript 3[49].

**SPR analysis.** SPR analysis was performed using Biacore T200 (GE healthcare) at 25 °C. Wild-type or mutant His$_6$-yNpl4$^{113-580}$, yUfd1–His$_6$-yNpl4$^{113-580}$, or His$_6$-hNpl4$^{105-600}$ was immobilized on a CM5 sensor chip by the amine-coupling method in 10 mM HEPES-NaOH (pH 7.5) containing 150 mM NaCl and 0.05% Tween-20. The amount of the immobilized ligand for each experiment is shown in response units (RU) in Supplementary Fig. 7. Ub$_2$ or Ub$_4$ was prepared in a two-fold serial dilution series and each dilution sample was injected for 60 s at a flow rate of 10 μL per min in 10 mM HEPES-NaOH buffer (pH 7.5) containing 150 mM NaCl, 0.05% Tween-20, and 0.5 g L$^{-1}$ BSA. The ranges of the concentrations of Ub$_2$ or Ub$_4$ were shown in Supplementary Fig. 7. Equilibrium dissociation constants ($K_d$) were calculated using Biacore T200 software. Data are shown as means ± standard deviation from three independent experiments for each sample.

**Fluorescence anisotropy-based affinity measurement.** The purified His$_6$-SUMO-yUfd1$^{288-305}$-GWCCPGCC was labeled with an equimolar ratio of FlAsH-EDT2 (Santa Cruz) in 50 mM Tris-HCl buffer (pH 8.0) containing 0.1% β-mercaptoethanol at room temperature for 1 h. Unreacted FlAsH-EDT2 was removed with PD-10 desalting columns (GE Healthcare). To measure binding affinities ($K_d$) of yNpl4$^{113-580}$ for yUfd1$^{288-305}$, 10 μL of 1 nM FlAsH-labeled His$_6$-SUMO-yUfd1$^{280-305}$ was aliquoted into a 384-well black, low volume, and round bottom plate (Corning). His$_6$-yNpl4$^{113-580}$ was prepared in a two-fold serial dilution series. Ten microliter of each diluted sample was added to wells containing FlAsH-labeled His$_6$-SUMO-yUfd1$^{280-305}$. Fluorescence polarization was recorded by an EnVision plate reader (PerkinElmer) using $\lambda_{ex} = 480$ nm and $\lambda_{em} = 535$ nm at 25 °C. $K_d$ was calculated with the program KaleidaGraph (HULINKS) (Supplementary Fig. 8). The measurement was carried out three times for each sample in 50 mM Tris-HCl buffer (pH 8.0) containing 100 mM NaCl, 5 mM β-mercaptoethanol, and 0.1 g L$^{-1}$ BSA.

**Yeast strains and media.** *S. cerevisiae* strains used in this study are listed in Supplementary Table 2. All strains are isogenic to W303. Unless otherwise stated, we used exponentially growing yeast cells cultured at 25 °C in YPD medium (1% yeast extract, 2% peptone, 2% glucose, 400 mg L$^{-1}$ adenine sulfate, and 10 mg L$^{-1}$ uracil) below an OD$_{600}$ of 0.8 (OD$_{600}$ of 1 contained $3.80 \times 10^7$ cells mL$^{-1}$ in our culture).

**Immunoblotting of yeast total cell lysate.** Cells corresponding to 1 OD$_{600}$ were harvested and extracted by the mild-alkali method[50]. Proteins were separated by SDS-PAGE on 4–12% NuPAGE Bis-Tris gels (Life Technologies) with MES buffer (Thermo Fisher Scientific) and transferred to PVDF membrane (GE Healthcare) on an XCell II Blot Module (Life Technologies). Immunoblotting was performed with the following antibodies: rabbit monoclonal antibody against K48 chains (Apu2; used at 1:1000 for immunoblotting; Millipore, Cat#05-1307), mouse monoclonal antibody against the FLAG-tag (M2, HRP-conjugated; 1:2000; Sigma-Aldrich, Cat #A8592), rabbit polyclonal antibody against Cdc48 (1:1000; gift from Dr. Kimura) and mouse monoclonal antibody against Pgk1 (22C5D8; 1:1000; Life Technologies, Cat#459250). HRP-conjugated goat anti-mouse or -rabbit Ig (1:10,000), used as a secondary antibody, was purchased from Jackson ImmunoResearch Laboratories (Cat#315-035-048 or Cat# 111-035-144). Immunoblots were developed using ECL

Prime Western Blotting Detection Reagent (GE Healthcare) and analyzed on ImageQuant LAS4000 (GE Healthcare).

**Yeast plate test.** Yeast cells were grown in SC medium (0.67% yeast nitrogen base without amino acids, 0.5% casamino acids, 2% glucose, 10 mM potassium phosphate (pH 7.5), 400 mg L$^{-1}$ adenine sulfate, 10 mg L$^{-1}$ uracil, and 20 mg L$^{-1}$ tryptophan) at 25 °C. Then serial dilutions (1:5) of overnight cultures were spotted onto SC agar plates.

**ATPase assay.** Solution (40 μL) containing 150 nM Cdc48, 150 nM UN hetero-dimer and 150 nM Ub$_n$-GFP substrate in reaction buffer (50 mM HEPES-NaOH (pH 7.5), 50 mM NaCl, 10 mM MgCl$_2$, 0.5 mM TCEP, 0.1 mg mL$^{-1}$ BSA) were incubated at 37 °C for 15 min. To this, 10 μL of 1 mM ATP solution and 50 μL of BIOMOL Green were added. Absorbance (600 nm) was measured at 5 s intervals in Enspire2300 (Perkin Elmer). The slope of the initial (0 to 50 s), linear portion of the curve was used to calculate the rate. The data were normalized to the average of the ATP hydrolysis rates of the wild-type Cdc48–UN complex without substrate.

**GST pulldown assay.** For analysis of the Cdc48 complex formation, GST-Npl4 or GST-Ufd1 (0.5 μM) was immobilized on 5 μL of Glutathione Sepharose 4B beads (GE Hearthcare) and incubated for 1 h at room temperature with 0.5 μM binding partner (untagged Ufd1 or His$_6$-tagged Npl4) and 3 μM Cdc48 (0.5 μM hexamer) in 100 μL of 50 mM Tris-HCl (pH 7.5) buffer containing 100 mM NaCl, 10% glycerol, and 0.1% Triton X-100 (binding buffer). For analysis of Ub chain binding, 1 μg of polyUb chains were added in 100 μL of the binding buffer. For analysis of Ub chain binding with the increasing concentrations of yUfd1, 0.5 μg of polyUb chains and 0, 0.25, 0.5, 1, 2, or 4 μM yUfd1 were added in 50 μL of the binding buffer. After three washes with the binding buffer, the bound proteins were eluted with NuPAGE LDS sample buffer (Thermo Fisher Scientific) for 10 min at 70 °C. The eluted proteins were separated by SDS-PAGE on 4–12% NuPAGE Bis-Tris gels and visualized with Bio-Safe Coomassie Stain (Bio-Rad). Immunoblots were performed as described above with mouse monoclonal antibody against Ub (P4D1, HRP-conjugated; 1:500; Santa Cruz Biotechnology # sc-8017 HRP).

**Immunoprecipitation of 3xFLAG-tagged NPL4 proteins.** Cell lysis and immunoprecipitation were performed essentially as previously described[15]. For anti-FLAG immunoprecipitation, 5 μL of anti-DDDDK-tag mAb-Magnetic beads (MBL M185-11) were used to precipitate FLAG-tagged protein complex from 1 mg of cell lysate by incubating for 1 h at 4 °C. After three washes with buffer A (50 mM Tris-HCl (pH 7.5), 100 mM NaCl, 10% glycerol, 10 μM bortezomib (LC Laboratories), 10 mM iodoacetamide, and 1x Complete protease inhibitor cocktail (EDTA free; Roche)) containing 1% Triton X-100, the bound proteins were eluted with 1X NuPAGE LDS sample buffer for 10 min at 70 °C.

**Reporting summary.** Further information on research design is available in the Nature Research Reporting Summary linked to this article.

## Data availability

The coordinates and structure factors of yNpl4, yNpl4–K48-Ub$_2$, and yNpl4–yUfd1 have been deposited in the Protein Data Bank under the accession codes 6JWH [https://doi.org/10.2210/pdb6jwh/pdb], 6JWI [https://doi.org/10.2210/pdb6jwi/pdb], and 6JWJ [https://doi.org/10.2210/pdb6jwj/pdb], respectively. The uncropped gel and blot images for Figs. 2d, 3e–f, and 4a–b and Supplementary Figs. 1d, 3a–d, 5a–e, and 6c are provided as a Source Data file. Other data are available from the corresponding authors upon reasonable request.

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

## Acknowledgements
We thank the beamline staff of the biological crystallography beamlines of BL41XU of SPring-8 (Hyogo, Japan) for technical help during data collection. This work was supported by JSPS/MEXT KAKENHI (JP16H04750 to Y. Sato, JP18K14913 to H.T., JP18H05498 to Y. Saeki, and JP18H05501 to S.F.), and Takeda Science Foundation (to Y. Saeki and K.T.).

## Author contributions
The study was conceived by Y. Sato, Y. Saeki and S.F.Y. Sato carried out biophysical experiments and structure determination and wrote the paper. H.T. and Y. Saeki performed yeast experiments. A.Y. and K.O. assisted with the structure determination. K.T. supervised the yeast experiments. All authors discussed the results. S.F. and Y. Saeki supervised the work, designed the experiments and wrote the paper.

## Competing interests
The authors declare no competing interests.
