## [Peer Review File · Nature Communications]

Reviewers' comments:

Reviewer #1 (Remarks to the Author):

In this contribution Sato et al. report crystal structures of a large fragment of yeast Npl4 in its apo-state, in complex with K48-linked di-ubiquitin and in complex with a short peptide derived from Ufd1. The latter two structures represent the most novel structural aspects of this study. The authors also use biochemical data (qualitative and quantitative binding studies) to corroborate the structural data. In its present form there are major weaknesses of the manuscript as described in detail below.

Major points:

1. Biochemical studies:

- a. It appears as if all binding studies were conducted with one protein batch, i.e. the number of biological replicates is equal to one. At a minimum a second protein batch has to be used for all biochemical studies.
- b. I fail to see how a relative activity can be negative (Fig. 2e), 0% relative activity should be the absolute lower limit.
- c. To put the affinities of Npl4 to ubiquitin chains into perspective, please carry out measurements with the Ufd1-Npl4 complex. I would expect tighter binding for the complex, even though the authors failed to measure binding of the isolated Ufd1 protein.
- d. According to the model (Fig. 6d) the ubiquitin moieties at positions 3 and 4 should also be in contact with Npl4 (CTD for Ub at position 3 and NBM for Ub at position 4). Please provide biochemical data which support this model. Please describe what is shown here for the D1 and D2 rings of *C. thermophilum* Cdc48. Is this a map, if yes, what is the contour level, or is it a surface representation of the model shown in Fig. 6a? The motivation for this comment is the width of the pore in the D1 ring, which seems surprisingly wide.
- e. Please generate the M574A variant and measure its activity in addition to the M547Q variant to match the character (Ala-scanning) of the T571A and I575A variants.
- f. For Table 3, check binding of the L292A variant. For the G297 variants, what are the backbone dihedral angles of G297, i.e. would they be accessible for non-Gly residues?
- g. Please define what is meant by "20% input" in Figs. 2d, 3e, 3f, 4a.

2. Crystallographic data: Obviously the density for the distal ubiquitin is very weak and the possible G76-K48 link (Fig. 2C) cannot be taken to indicate that its position is correct. Unfortunately, the biochemical data also do not support that it was correctly modeled. Please provide evidence (structural or biochemical) that the distal ubiquitin really binds as indicated. The anomalous peak (Fig. S1c) somewhere between Met1 and Met19 does not really support the assignment. Playing devil's advocate, maybe it only corresponds to either Met1 or Met19 and hence the distal ubiquitin needs to be rotated, which could also shorten the unrealistically long G76-K48 isopeptide bond.

a. All 2F(o)-F(c) electron density maps have to be replaced with omit maps. For highly mobile regions it might be quite ok to use a lower contour level. As long as the density only appears around the model (in contrast to everywhere as one would expect for noise), it represents true features.

b. Table 1: Add R(pim), CC(1/2) and Ramachandran statistics for each structure. I/sigI should read <I/sigI>.

c. I strongly disagree with the high resolution limits in each case. The <I/sigI> > 2 criterion is outdated and the high resolution limit should be set based to where CC(1/2) drops below 0.5 (possibly even below 0.3). Correspondingly the data have to be reprocessed to higher resolution and the structures re-refined.

d. Please describe whether there is/are one or more copies of a given structure in the asymmetric unit. If there is more than one (presumably that is the case for the di-ubiquitin complex), indicate whether ncs restraints were used during refinement or not. If that is the case, was twofold ncs averaging employed to improve the maps of the bound Ub molecules.

e. Experimental section: Please indicate the wavelengths for each dataset, detector type and temperature during data collection.

3. Presentation of data (see also below)

a. The figure legends are inadequate, i.e. do not provide the reader with the necessary information to understand the figures.

b. Some of the figures are very difficult to comprehend. Fig. 2c has poor contrast between model and maps (especially blue trace and blue density). Fig. 3e and 4f: Obviously activities are represented relative to WT (which is set to 1), however, are we to believe that there is no variation in the WT measurements? Please also indicate the standard deviations for the wild-type measurements.

Minor points:

1. Define all abbreviations used in the abstract or try to avoid using any abbreviations in the abstract.

2. Abstract, line 4: Replace "domain" with "construct" or similar.

3. Page 3, line 5 from bottom: The participation of a DUB has been questioned, see reference 7 from your list of references (which should be cited here).

4. Page 3, line 3 from bottom: Please use "UBXL domain" throughout and either use mNpl4 or NPLOC4 consistently throughout the text.

5. Fig. 3B: Add labels for Leu292 and Pro295

6. Page 10, title: Replace "Ufd1 are not overlapped with" with "Ufd1 do not overlap with". Same in line 5 on page 11.

7. Page 11, line 6 from bottom: Tone down statement , i.e. "is require" since residual binding is left.

8. Page 12, lines 2 and 3 from bottom: Replace "These findings suggest the C-terminal" with "These finding suggests that the C-terminal".

9. Page 13, line 5: Replace "the C-terminal of the full-length" with "the C-terminal end of the full-length"

10. Page 14/15: Replace "The present yUfd1(288-305) model can fit into the unassigned density observed in the cryo-EM of the Cdc48-UN complex" with "The present yUfd1(288-305) model can be fitted into the unassigned density observed in the cryo-EM structure of the Cdc48-UN complex".

11. Legend to Figure 2:
 - (a) Add "in two orientations"
 - (b) Orange dotted lines are barely visible.
 - (c) Where is the isopeptide linkage supposed to be? Clearly there is no density supporting its existence, at the same time the black line from the Lys48 to Arg72 is not really relevant as the isopeptide bond should connect to Gly76.

12. Legend to Figure 3:

(a) Add "in two orientations" and "H-bonds are shown as ..."

13. Figure 5:

Include the contour level of the cryo-EM map

(a) Add "in two orientations"

14. Figure S4:

The conserved amino acids (white on red) are barely readable. Please increase contrast.

15. Figure S6:

(a) Add info in legend that the "down conformation" is shown on the left and the "up conformation" is on the right, even though this info is in the figure.

16. Replace "plante" throughout the manuscript with "plantae" as this refers to the kingdom of plants.

17. Page 23, middle: In the Cdc48 purification there is contradictory info regarding the TCEP concentration: Is it 1 mM or 0.5 mM?

18. Please define ND as "no binding detected" in Table 2. Presumably it does not mean that data were "not determined" as in not measured.

19. Fig. 3g: What does the weak band between the Cdc48 and FLAG signals correspond to? In my printout there is a rectangular box with a cross in it at this position. Is that supposed to be a label for this band, if yes, what does it tell the reader?

20. Fig. S6: If I understand this figure correctly, the authors want to show that the UBXL, SHP1 and SHP2 elements contact three separate Cdc48 protomers at what looks to me at positions I (UBXL), III (SHP1) and IV (SHP2) of the Cdc48 hexamer. The lengths of the linker segments should be converted

from a.a. into A-values to allow for a direct comparison with the figure. The strain aspect of this model (Fig. 6b) is not obvious from the distances, at most the NBM-SHP2 linker would be too short to reach the N domain of protomer IV (by the way it looks as if the distance would be shorter to the N domain in protomer V), however, whether SHP2 actually interacts with Cdc48 has not been established. Overall it seems that this model is too speculative.

Reviewer #2 (Remarks to the Author):

How yeast Npl4 recognizes polyubiquitin is not fully understood. In this manuscript from Sato et al, the authors determine and compare crystal structures of yeast Npl4 (γ Npl4) encompassing the Zinc finger, MPN and CTD domains in complex with Lys48-linked diubiquitin and with Ufd1. While the MPN groove was expected to bind ubiquitin, the authors instead find that Ufd1 binds to this region. Ubiquitin interactions with distinct regions in the Npl4-CTD enable binding to K48-Ub2. The authors show that proximal ubiquitin interaction is important for linkage-selectivity. Further, structure-guided mutations disrupt ubiquitin binding and stimulation of the ATPase activity of Cdc48-Ufd1-Npl4 complex. This work therefore gives insights into polyubiquitin binding and offers a model for the assembly and working of this complex.

Comments:

1. On page 5, the authors state that molecular replacement was carried out using 1UBQ as the search model, which did not provide a solution. However, residual electron density corresponding to K48-Ub2 was found. Does this mean that the authors use K48-Ub2 as a search model? Moreover, electron density for the proximal Ub is very weak. One possibility is that ubiquitin is binding to Npl4 in multiple conformations possibly due to the weak interaction interface of the proximal Ub. Refining the structure at lower resolution may help resolve this, and identify conformers that bind with lower occupancy.
2. The binding mode of the C-terminal helix in CTD is compared to being similar to that of the UMI domain from RNF168. Please superpose these structures and discuss how exactly the binding modes are similar. Also show how this differs from UIM/MIU interactions.
3. On page 7, the authors state that affinity for K48 Ub2 is too low whereas higher affinity was observed with Ub4. Does this mean that there are additional ubiquitin binding sites/
4. The presentation of results in page 6-7 is a bit confusing and jumbled. For instance, part of the result shown in Figure 2d is already mentioned in the previous section on Page 6. But only in page 7 do the authors describe the experimental approach.
5. the authors do not offer any explanation for why their results on Ufd1 binding to ubiquitin chains differ or contradict previously published data in REFs 23, 29.

6. could the authors state and comment on the observation that more polyubiquitin is bound by Npl4 in the presence of Cdc48
7. The interaction of ubiquitin with MPN domain is not described in detail. Further, mutations of MPN to analyse the contribution of the MPN to polyubiquitin binding would strengthen this part of the study.
8. It is unclear why the authors did not mutate S498 to a larger charged residue to repel interaction with the proximal ubiquitin. Rather they insert loops which may perturb local structure.
9. On page 9, the authors state that A494F mutation “shows a negligible effect on affinity”. However, they contradict themselves as in the previous page they make a point of how this mutation increases affinity.
10. The authors find that γ Ufd1 does not inhibit binding of γ Npl4 to ubiquitin chains. Only one concentration of γ Ufd1 was tested and I would suggest testing increasing concentrations of γ Ufd1 on Npl4-polyubiquitin interaction.
11. The discussion section is confusing to read and I would suggest that the authors restructure and rewrite this section to make it more understandable. Further, some of the figures that they base their discussion on are in supplementary information. I would suggest moving these into main figures to make it easier to read.

Re: NCOMMS-19-15801-T

We are grateful for the reviewers' comments and made best efforts to address them. We believe that the changes based on the comments have greatly improved the manuscript.

Comments from Reviewer #1

Major points:

1. Biochemical studies:

a. It appears as if all binding studies were conducted with one protein batch, i.e. the number of biological replicates is equal to one. At a minimum a second protein batch has to be used for all biochemical studies.

For SPR analysis (Table2) and fluorescence polarization-based analysis (Table3), wild-type yNpl4 and yUfd1-FIAsH were purified again, and the experiments were repeated. We confirmed that the activity differences between different batches were negligible. For GST pulldown assay, all experiments were repeated twice with two different batches of samples.

Fig. 2d; Supplementary Fig. 3a

Fig. 3e; Supplementary Fig. 5a, b

Fig. 3f; Supplementary Fig. 5d

Fig. 4a, b; Supplementary Fig. 6c

Supplementary Fig. 3b

Supplementary Fig. 3c

Supplementary Fig. 3d

Supplementary Fig. 5c

b. I fail to see how a relative activity can be negative (Fig. 2e), 0% relative activity should be the absolute lower limit.

In the time course experiment in the previous Fig. 2e, a few of the samples with almost no ATPase activity showed apparently decreased signals, which were interpreted as negative activity. We performed the experiment again. At this time, no samples showed negative activity (Fig. 2e). One representative plot (out of ten technical replicates) is shown in Supplementary Fig. 3e.

c. To put the affinities of Npl4 to ubiquitin chains into perspective, please carry out measurements with the Ufd1-Npl4 complex. I would expect tighter binding for the complex, even though the

authors failed to measure binding of the isolated Ufd1 protein.

To analyze the binding affinity between the Ufd1–Npl4 complex and K48-Ub₄, SPR analysis using the purified yUfd1–yNpl4¹¹³⁻⁵⁸⁰ complex and K48-Ub₄ was performed. The result showed that the presence of yUfd1 has little effect on the affinity of yNpl4 for K48-Ub₄ (Table 2). We also tested the effect of increasing concentrations of yUfd1 on the interaction between yNpl4 and K48-linked Ub chains (Supplementary Fig. 3d). The result showed that yUfd1 hardly affects the binding activity of yNpl to K48-linked Ub chains. These were described in page 8, lines 3-6.

d. According to the model (Fig. 6d) the ubiquitin moieties at positions 3 and 4 should also be in contact with Npl4 (CTD for Ub at position 3 and NBM for Ub at position 4). Please provide biochemical data which support this model. Please describe what is shown here for the D1 and D2 rings of C. thermophilum Cdc48. Is this a map, if yes, what is the contour level, or is it a surface representation of the model shown in Fig. 6a? The motivation for this comment is the width of the pore in the D1 ring, which seems surprisingly wide.

During the revision of this manuscript, a cryo-EM structure of the substrate-engaged Cdc48–UN complex was reported (Twomey, et al., 2019). The comparison of our structures with the substrate-engaged Cdc48-UN structure was shown in Fig. 6, which was substituted for the comparison with Cdc48-UN in the previous Fig. 5 (“Fig. 6” in this comment likely means “Fig. 5”, judging from the context). The cryo-EM map of the substrate-engaged Cdc48–UN complex was shown in Fig. 6c with the information of the contour level of the map in the figure legend.

e. Please generate the M574A variant and measure its activity in addition to the M547Q variant to match the character (Ala-scanning) of the T571A and I575A variants.

The yNpl4 M574A variant was examined by SPR analysis (Table 2), GST pulldown assay (Fig. 2d and Supplementary Fig. 3a, 3b), co-immunoprecipitation of ubiquitylated proteins in yeast (Supplementary Fig. 3c), and ATPase assay (Fig. 2e). The results indicated that the effect of M574A is milder than those of T571A, M574Q, and I575A.

f. For Table 3, check binding of the L292A variant. For the G297 variants, what are the backbone dihedral angles of G297, i.e. would they be accessible for non-Gly residues?

Binding between the yUfd1 L292A variant and yNpl4 was analyzed. The result was added in Table 3. The backbone dihedral angles ϕ and ψ of Gly297 are 90.1° and -8.5° , respectively. MolProbity, a standard program for protein structure validation, judges these angles as “favored” for Gly but “allowed” for non-Gly residues. Gly is more favorable than non-Gly residues at residue 297. This was described in line 26 (the last line), page10–lines 1-3, page 11.

g. Please define what is meant by "20% input" in Figs. 2d, 3e, 3f, 4a.

Definition of "20% input" was described in each figure legend. Briefly, “20% input” means 20% of the volume of each sample that was applied to affinity resin.

2. Crystallographic data:

Obviously the density for the distal ubiquitin is very weak and the possible G76-K48 link (Fig. 2C) cannot be taken to indicate that its position is correct. Unfortunately, the biochemical data also do not support that it was correctly modeled. Please provide evidence (structural or biochemical) that the distal ubiquitin really binds as indicated.

The anomalous peak (Fig. S1c) somewhere between Met1 and Met19 does not really support the assignment. Playing devil's advocate, maybe it only corresponds to either Met1 or Met19 and hence the distal ubiquitin needs to rotated, which could also shorten the unrealistically long G76-K48 isopeptide bond.

The density of the proximal Ub (but not the distal Ub) is actually weak. We reprocessed the diffraction data according to the strategy suggested in the comment 2c of Reviewer #1. As a result, the electron density of Arg42 of the proximal Ub was observed in $2F_o - F_c$ map, $F_o - F_c$ omit map (calculated with the proximal Ub removed) (Supplementary Fig. 1b), and $2F_o - F_c$ composite omit map. Ser498 of yNpl4 forms a hydrogen bond with Arg42 of the proximal Ub in our structure (Fig. 2b). Therefore, the yNpl4 S498R variant was examined by GST pulldown assay, SPR analysis, and ATPase assay. The GST pulldown assay showed that the S498R mutation of yNpl4 decreased the binding to K48-linked Ub chains (Fig. 2d and Supplementary Fig. 3b). The ATPase assay showed that this mutation decreased the ATPase activity of the Cdc48-UN complex (Fig. 2e). The SPR analysis showed that the S498R mutation of yNpl4 decreased the binding affinity to K48-Ub₄ but not to K63- or M1-Ub₄ (Table 2). These results are consistent with the finding that Ser498 of yNpl4 is located on the binding surface of K48-linked Ub chains. Furthermore, we also observed the anomalous peak of yNpl4-K48-Ub₂ (I30M) in addition to those of the P19M and V26M mutants. Although the anomalous peaks derived from SeMet30 of the distal Ub and SeMet1 of the

proximal Ub were not detected, those from SeMet1 of the distal Ub and SeMet30 of the proximal Ub were detected (Supplementary Fig. 1c). Taken together, we could confirm the positions of Met1, Met19, and Met26 of the distal Ub and Met1, Met26, and Met30 of the proximal Ub. The position and orientation of Ub are uniquely determined by the location of three points that are not colinear. We therefore present structural and biochemical evidences to prove that the proximal Ub really binds as indicated.

a. All $2F_o-F_c$ electron density maps have to be replaced with omit maps. For highly mobile regions it might be quite ok to use a lower contour level. As long as the density only appears around the model (in contrast to everywhere as one would expect for noise), it represents true features.

$2F_o-F_c$ maps in Fig. 2c and Supplementary Fig. 1 were replaced with $2F_o-F_c$ composite omit map and F_o-F_c omit map (calculated with the distal or proximal Ub removed), respectively.

b. Table 1: Add $R(\text{pim})$, $CC(1/2)$ and Ramachandran statistics for each structure. $I/\sigma I$ should read.

$R(\text{pim})$, $CC(1/2)$, and Ramachandran statistics for each structure were added in Table 1.

c. I strongly disagree with the high resolution limits in each case. The > 2 criterion is outdated and the high resolution limit should be set based to where $CC(1/2)$ drops below 0.5 (possibly even below 0.3). Correspondingly the data have to be reprocessed to higher resolution and the structures re-refined.

According to this comment, we reprocessed the data and re-refined the structures of yNpl4–K48Ub₂ and yNpl4–yUfd1 to higher resolutions (Table 1). The resolution of the data of yNpl4 alone was limited by the setting of the data collection (*i.e.*, detector size and the distance between the detector and the crystal) and therefore could not be extended.

d. Please describe whether there is/are one or more copies of a given structure in the asymmetric unit. If there is more than one (presumably that is the case for the di-ubiquitin complex), indicate whether ncs restraints were used during refinement or not. If that is the case, was twofold ncs averaging employed to improve the maps of the bound Ub molecules.

There is one copy of the structure in the asymmetric unit of the crystals of yNpl4 alone and the yNpl4-yUfd1 complex. On the other hand, there are two yNpl4 and two Ub molecules in the asymmetric unit of the yNpl4-K48-Ub₂ crystal. NCS restraints were applied during refinement of the yNpl4-K48-Ub₂ structure and improved the R_{free} value, but had little effect on the map quality of the Ub moieties. The number of copies in the asymmetric unit and the NCS refinement were described in the Method section.

e. Experimental section: Please indicate the wavelengths for each dataset, detector type and temperature during data collection.

The wavelength, detector type, and temperature during data collection for each dataset were described in the Method section.

3. Presentation of data (see also below)

a. The figure legends are inadequate, i.e. do not provide the reader with the necessary information to understand the figures.

The figure legends were revised accordingly.

b. Some of the figures are very difficult to comprehend.

Fig. 2c has poor contrast between model and maps (especially blue trace and blue density).

The $2F_o-F_c$ map was replaced with a $2F_o-F_c$ composite omit map according to the comment 2a of Reviewer #1. The color of the map was changed from blue to olive to improve the contrast.

Fig. 3e and 4f: *Obviously activities are represented relative to WT (which is set to 1), however, are we to believe that there is no variation in the WT measurements? Please also indicate the standard deviations for the wild-type measurements.*

We performed ATPase assays in the presence and absence of substrates (K48-Ubn-GFP), and normalized the data so that the ATPase activity of the substrate-free sample was set to 1 for each experiment in the initial manuscript. Therefore, the activities of the substrate-free samples were always 1, and there was no variation in the previous Fig. 2e and 4b (“Fig. 3e”

and “Fig. 4F” in this comment may be “Fig. 2e” and “Fig. 4b”, respectively, judging from the context).

In the revised manuscript, the average of ATPase activity of the substrate-free sample was set to 1. In the graphs presenting the results of the ATP assays, all data are plotted as dots to show the variation of the data, instead of indicating the standard deviations (Fig. 2e and 4c).

Minor points:

1. Define all abbreviations used in the abstract or try to avoid using any abbreviations in the abstract.

The abstract was revised accordingly.

2. Abstract, line 4: Replace "domain" with "construct" or similar.

"zf-Npl4-MPN-CTD domain" was removed in the abstract, line 4, according to the minor comment 1 of Reviewer #1.

3. Page 3, line 5 from bottom: The participation of a DUB has been questioned, see reference 7 from your list of references (which should be cited here).

The description about the participation of a DUB was removed.

4. Page 3, line 3 from bottom: Please use "UBXL domain" throughout and either use mNpl4 or NPLOC4 consistently throughout the text.

We used the terms "UBXL domain" and “hNpl4” consistently throughout the text.

5. Fig. 3B: Add labels for Leu292 and Pro295

Labels were added for Leu292 and Pro295 of yUfd1 in Fig. 3b.

6. Page 10, title: Replace "Ufd1 are not overlapped with" with "Ufd1 do not overlap with". Same in line 5 on page 11.

This phrase was revised accordingly.

7. Page 11, line 6 from bottom: Tone down statement, i.e. "is require" since residual binding is left.

This state statement was toned down as follows:

“Thus, the UN heterodimer formation facilitates the Cdc48–UN assembly, although both yNpl4 and yUfd1 directly bind to Cdc48.”

8. Page 12, lines 2 and 3 from bottom: Replace "These findings suggest the C-terminal" with "These finding suggests that the C-terminal".

This phrase was revised accordingly (i.e., “that” was inserted).

9. Page 13, line 5: Replace "the C-terminal of the full-length" with "the C-terminal end of the full-length"

This phrase was revised accordingly.

10. Page 14/15: Replace "The present yUfd1(288-305) model can fit into the unassigned density observed in the cryo-EM of the Cdc48-UN complex" with "The present yUfd1(288-305) model can be fitted into the unassigned density observed in the cryo-EM structure of the Cdc48-UN complex".

This sentence was revised as follows:

"The present yUfd1^{288–305} model was fitted into the unassigned density observed in the cryo-EM map,"

II. Legend to Figure 2:

(a) Add "in two orientations"

This phrase was revised accordingly.

(b) Orange dotted lines are barely visible.

The orange dotted lines were replaced with black ones. In addition, the model was slightly rotated to clearly show the hydrogen bonds between yNpl4 and Ub.

(c) Where is the isopeptide linkage supposed to be? Clearly there is no density supporting its existence, at the same time the black line from the Lys48 to Arg72 is not really relevant as the isopeptide bond should connect to Gly76.

We reprocessed the diffraction data according to the strategy suggested in the comment 2c of Reviewer #1. As a result, the electron density of Arg72 of the distal Ub was diminished in $2F_o - F_c$ map, $F_o - F_c$ omit map (calculated with the distal Ub removed), and $2F_o - F_c$ composite omit map. Therefore, we removed the Arg72 of the distal Ub from the model of yNpl4-K48-Ub₂. The isopeptide bond of Ub chains is flexible, even in complex with Ub chain-binding domains, as shown in previous crystallographic studies. The electron density of the isopeptide bond is invisible in the crystal structures of Mindy-1-K48-Ub₂ and AIRAPL-K48-Ub₃. In such cases, the possibility of the Ub-Ub linkage can be assessed by calculating the C α -C α distance between Leu71 of the distal Ub and Lys48 of the proximal Ub. The range of this distance should be 9–20 Å to enable the isopeptide bond.

In our yNpl4-K48-Ub₂ structure, this distance is 11 Å. Furthermore, we examined the yNpl4-K48-Ub₂ crystals by SDS-PAGE (Supplementary Fig. 1d) and confirmed that K48-Ub₂ was retained in the crystals. Therefore, we concluded that the isopeptide bond connects the distal and proximal Ub moieties in the present yNpl4-K48-Ub₂ structure.

12. Legend to Figure 3:

(a) Add "in two orientations" and "H-bonds are shown as ..."

These phrases were added in the legend of Fig. 3 accordingly.

13. Figure 5:

Include the contour level of the cryo-EM map

During the revision of this manuscript, a cryo-EM structure of the substrate-engaged Cdc48-UN complex was reported (Twomey, et al., 2019). The comparison of our structures with the substrate-engaged Cdc48-UN structure was shown in Fig. 6, which was substituted for the comparison with Cdc48-UN in the previous Fig. 5. The cryo-EM map of Cdc48-UN (EMD-7476) was replaced with that of the substrate-engaged Cdc48-UN (EMD-0665). The

contour level of the cryo-EM map was indicated in Fig. 6.

(a) Add "in two orientations"

The previous Fig. 5 was removed in the revised manuscript.

14. Figure S4:

The conserved amino acids (white on red) are barely readable. Please increase contrast.

The conserved amino acids were indicated by white characters with black backgrounds to increase the contrast.

15. Figure S6:

(a) Add info in legend that the "down conformation" is shown on the left and the "up conformation" is on the right, even though this info is in the figure.

Supplementary Fig. 6 was removed, according to the minor comment 20 of Reviewer #1.

16. Replace "plante" throughout the manuscript with "plantae" as this refers to the kingdom of plants.

This typo was corrected accordingly.

17. Page 23, middle: In the Cdc48 purification there is contradictory info regarding the TCEP concentration: Is it 1 mM or 0.5 mM?

The TCEP concentration is 1 mM. This typo was corrected.

18. Please define ND as "no binding detected" in Table 2. Presumably it does not mean that data were "not determined" as in not measured.

In the previous Table 2, ND meant the case that the actual K_d value is above half of the upper limit of the substrate concentration used in the experiment. In the new Table 2, K_d values that may be underestimated were shown in parentheses.

19. *Fig. 3g: What does the weak band between the Cdc48 and FLAG signals correspond to? In my printout there is a rectangular box with a cross in it at this position. Is that supposed to be a label for this band, if yes, what does it tell the reader?*

Fig. 3g was moved to Supplementary Fig. 5e. The weak band corresponds to a non-specific band. This was described in the legend.

20. *Fig. S6: If I understand this figure correctly, the authors want to show that the UBXL, SHP1 and SHP2 elements contact three separate Cdc48 protomers at what looks to me at positions I (UBXL), III (SHP1) and IV (SHP2) of the Cdc48 hexamer. The lengths of the linker segments should be converted from a.a. into Å-values to allow for a direct comparison with the figure. The strain aspect of this model (Fig. 6b) is not obvious from the distances, at most the NBM-SHP2 linker would be too short to reach the N domain of protomer IV (by the way it looks as if the distance would be shorter to the N domain in protomer V), however, whether SHP2 actually interacts with Cdc48 has not been established. Overall it seems that this model is too speculative.*

Supplementary Fig. 6 was removed accordingly.

Comments from Reviewer #2

1. *On page 5, the authors state that molecular replacement was carried out using 1UBQ as the search model, which did not provide a solution. However, residual electron density corresponding to K48-Ub₂ was found. Does this mean that the authors use K48-Ub₂ as a search model?*

The linkage of Ub chains (*i.e.*, the C-terminal tail of the distal Ub conjugated to the lysine residue of the proximal Ub) is flexible, and relative positions of the two Ub moieties in K48-Ub₂ differ in the reported apo and protein-bound structures of K48-linked Ub chains. Therefore, we used a crystal structure of monoUb (PDB: 1UBQ) as the search model. The description regarding the modeling of K48-Ub₂ was revised as follows:

"... Although molecular replacement using Ub (PDB 1UBQ) as the search model was unsuccessful, we found residual electron density corresponding to K48-Ub₂ and manually built the model of K48-Ub₂".

Moreover, electron density for the proximal Ub is very weak. One possibility is that ubiquitin is binding to Npl4 in multiple conformations possibly due to the weak interaction interface of the proximal Ub. Refining the structure at lower resolution may help resolve this, and identify conformers that bind with lower occupancy.

Based on this suggestion, we refined the structure at lower resolutions (3, 4, or 5 Å). However, the electron density of the proximal Ub was not improved, and other conformers could not be identified.

2. The binding mode of the C-terminal helix in CTD is compared to being similar to that of the UMI domain from RNF168. Please superpose these structures and discuss how exactly the binding modes are similar. Also show how this differs from UIM/MIU interactions.

The C-terminal helix of CTD is the primary Ub-binding site of yNpl4. Among the previously reported Ub-binding domains, UIM, MIU, and UMI have a single helix as a primary Ub-binding site. Npl4 CTD, MIU, and UMI but not UIM exhibit the same helix orientation relative to the bound Ub. Therefore, we compared Npl4 CTD, Rabex5 MIU, and RNF168 UMI (Fig. 5) and described similarity and difference of their structural features in the first paragraph of the Discussion section.

3. On page 7, the authors state that affinity for K48 Ub₂ is too low whereas higher affinity was observed with Ub₄. Does this mean that there are additional ubiquitin binding sites/

In the cryo-EM structure of the substrate-engaged Cdc48-UN complex, the unfolded Ub is bound to a site different from the K48-Ub₂-bound site observed in the present Npl4-K48-Ub₂ structure. However, this site seems unavailable for folded Ub. It still remains unclear whether additional binding sites for folded Ub exist or not in Npl4. Nevertheless, in general, longer Ub chains have higher affinity for Ub-binding proteins, even if the number of Ub moieties in the Ub chains is greater than the number of Ub-binding sites of the Ub-binding proteins. This point was discussed in lines 24-25, page 6.

4. The presentation of results in page 6-7 is a bit confusing and jumbled. For instance, part of the result shown in Figure 2d is already mentioned in the previous section on Page 6. But only in page 7 do the authors describe the experimental approach.

In this manuscript, we do not focus on the ZF domain of Npl4, and therefore removed the

description about it. Fig. 2d was not cited in Page 6 in the revised manuscript.

5. the authors do not offer any explanation for why their results on Ufd1 binding to ubiquitin chains differ or contradict previously published data in REFs 23, 29.

Ye *et al.*, *J. Cell Biol.*, 2003 and Park *et al.*, *Structure*, 2005 have shown that Ufd1 binds to K48-linked Ub chains through its UT3 domain. However, our pulldown analysis failed to detect the binding of yUfd1 to K48-linked Ub chains (Supplementary Fig. 3b). We next analyzed the binding between the GST-yNpl4 and K48-linked Ub chains on increasing concentrations of yUfd1 (0.25-4 μ M) by GST pulldown assay but did not observe significant effects (Supplementary Fig. 3d). SPR analysis using the purified yUfd1-yNpl4¹¹³⁻⁵⁸⁰ complex and K48-Ub₄ also showed that the presence of yUfd1 has little effect on the affinity of yNpl4 for K48-Ub₄ (Table 2). Although the K_d value of yUfd1 for polyUb has not been estimated, that for monoUb has been estimated to be within the range of 1-2 mM. This value is much higher than the K_d value of yNpl4 for K48-Ub₂ or K48-Ub₄ (Table 2), suggesting that the affinity of yUfd1 for K48 chains is much lower than that for yNpl4. Collectively, these *in vitro* and *in vivo* results suggest that the C-terminal helix of yNpl4 is the key determinant for Ub chain recognition by the Cdc48-UN complex. These were described in lines 1-11, page 8.

6. could the authors state and comment on the observation that more polyubiquitin is bound by Npl4 in the presence of Cdc48

We performed GST pulldown assays in the presence and absence of Cdc48 (Fig. 2d and Supplementary Fig. 3a, b). The results showed that Cdc48 has little effect on the binding affinity of yNpl4 to K48-linked Ub chains.

7. The interaction of ubiquitin with MPN domain is not described in detail. Further, mutations of MPN to analyse the contribution of the MPN to polyubiquitin binding would strengthen this part of the study.

In our structure, only His290 is involved in the interaction with the proximal Ub in the MPN domain. His290 interacts with the aliphatic portion of Thr9 of the proximal Ub. However, the electron density of Thr9 of the proximal Ub is ambiguous, and GST pulldown assay showed that the H290A mutation of yNpl4 has little effect on the binding between yNpl4 and K48-Ub₄. Therefore, we concluded that the CTD domain of yNpl4 is a primary binding site for K48-linked Ub chains.

8. It is unclear why the authors did not mutate S498 to a larger charged residue to repel interaction with the proximal ubiquitin. Rather they insert loops which may perturb local structure.

We generated the S498R and S498E mutants, and first tested their binding to K48-Ub₄ by GST pulldown assays. Since the result indicated that the S498R mutation was greater than the S498E mutation, the effect of the S498R mutation was further examined by SPR analysis and ATPase assay. The S498R mutation of yNpl4 decreased the binding to K48-linked Ub chains (Fig. 2d and Supplementary Fig. 3a, b) and the ATPase activity of the Cdc48-UN complex (Fig. 2e). Regarding the chain-type specificity, the S498R mutation of yNpl4 decreased the binding affinity for K48-Ub₄ but not for K63- or M1-Ub₄ (Table 2). These results are consistent with the finding that Ser498 of yNpl4 is located on the binding surface of the proximal Ub in K48-Ub₂.

9. On page 9, the authors state that A494F mutation “shows a negligible effect on affinity”. However, they contradict themselves as in the previous page they make a point of how this mutation increases affinity.

We described that the A494F mutation shows a negligible effect on affinity in the previous manuscript, because A494F mutation did not affect the ATPase activity of the Cdc48-UN complex even if A494F mutation slightly increased the affinity of Npl4 for K48-Ub₄. In this manuscript, we do not focus on the effect of the A494F mutation for the ATPase activity of the Cdc48-UN complex, and therefore removed this description.

10. The authors find that yUfd1 does not inhibit binding of yNpl4 to ubiquitin chains. Only one concentration of yUfd1 was tested and I would suggest testing increasing concentrations of yUfd1 on Npl4-polyubiquitin interaction.

We tested the effect of increasing concentrations of yUfd1 on the interaction between yNpl4 and K48-linked Ub chains (Supplementary Fig. 3d). This was described in lines 3-5, page 8.

11. The discussion section is confusing to read and I would suggest that the authors restructure and rewrite this section to make it more understandable. Further, some of the figures that they base their discussion on are in supplementary information. I would suggest moving these into main

figures to make it easier to read.

According to the comment of Reviewer #1, the discussion about the conformation of the Cdc48-UN complex (Supplementary Fig. 6 in the previous manuscript) was removed. On the other hand, according to the comment 2 of Reviewer #2, the comparison of the C-terminal helix of yNpl4 and other Ub-binding helices were added (Fig. 5). During the revision of this manuscript, a cryo-EM structure of the substrate-engaged Cdc48–UN complex was reported (Twomey, et al., 2019). The comparison of our structures with the substrate-engaged Cdc48-UN structure was shown in Fig. 6, which was substituted for the comparison with Cdc48-UN in the previous Fig. 5. All figures cited in the discussion section were included in the main figures.

REVIEWERS' COMMENTS:

Reviewer #1 (Remarks to the Author):

Sato et al. have substantially revised their initial manuscript including the addition of new experimental data as requested in the initial reviewing round. During revision of the manuscript the cryo-EM structure of a substrate-engaged p97-UN complex was published, however, this should according to my strong opinion should not preclude the publication of their work. The authors have taken advantage of this new info and compared their structure to the recent cryo-EM structure (Fig. 6) revealing that there are apparent differences in the location of the proximal ubiquitin moiety, which are interesting and presumably reflect the dynamic nature of the degradation process. I have a few minor comments regarding this figure. While panels a and b are straightforward to understand, I think panel c and its discussion in the text should be improved. (1) I find the contrast between the yellow ribbon and magenta density difficult to discern and for me the ribbon is barely visible. (2) Maybe the authors could show their UN structure and with a dashed box highlight the region of interest (as in panel a). (3) More importantly, what is the purpose of this figure and what is the significance of the unassigned residual density? Extra text in the manuscript is needed to provide the necessary information. At the same time the section about "inhibitors of the Ub-proteasome system" should be moved to a new paragraph.

Reviewer #2 (Remarks to the Author):

The authors have performed several new experiments and analyses that satisfactorily address my concerns and comments. The authors have also rewritten large parts of this manuscript and restructured the figures making this a much improved version.

This review was prepared by Yogesh Kulathu. To promote transparency and to accelerate scientific progress, I no longer participate in anonymous peer review. This review is provided with the understanding that my name will be available to the reviewers and authors

Re: NCOMMS-19-15801A

Comments from Reviewer #1

I have a few minor comments regarding this figure (Fig. 6). While panels a and b are straightforward to understand, I think panel c and its discussion in the text should be improved.

(1) I find the contrast between the yellow ribbon and magenta density difficult to discern and for me the ribbon is barely visible.

The color of the map was changed from magenta to grey to improve the contrast.

(2) Maybe the authors could show their UN structure and with a dashed box highlight the region of interest (as in panel a).

In the final version, the substrate-engaged Cdc48-bound UN structure with the cryo-EM density map was shown in Fig. 6c. The cartoon models of Npl4 and Cdc48 are derived from the cryo-EM analysis, whereas that of Ufd1 is derived from the present crystallographic analysis. The dashed box highlights the Ufd1 structure, and its close-up view is shown as Fig. 6d.

(3) More importantly, what is the purpose of this figure and what is the significance of the unassigned residual density? Extra text in the manuscript is needed to provide the necessary information. At the same time the section about "inhibitors of the Ub-proteasome system" should be moved to a new paragraph.

The Ufd1 structure from the present crystallographic analysis does not completely fill the unassigned density in the cryo-EM map of the substrate-engaged Cdc48–UN complex, and there remains residual density. The right panel of the previous Fig. 6c showed this residual density (incorrectly labeled as “unassigned density” in the previous version; sorry for this confusing mistake). In the final version, this panel was removed because we do not intend to highlight the residual density. Instead, the Ufd1 structure fitted into the cryo-EM map was clearly presented according to the comment (2). Concomitantly, the second last paragraph in the Discussion section was rewritten as follows:

“We also determined the crystal structure of yNpl4¹¹³⁻⁵⁸⁰–yUfd1²⁸⁸⁻³⁰⁵. We docked the structure of the NBM region of Ufd1 into the cryo-EM structure of the substrate-engaged

Cdc48–UN (Fig. 6c). The present yUfd1^{288–305} model was nicely fitted into the unassigned density observed in the cryo-EM map (Fig. 6c, d), which was supposed to correspond to a part of yUfd1. This indicates that the yNpl4–Ufd1 interaction revealed by crystallography of the UN complex similarly occurs in the substrate-engaged Cdc48–UN (Fig. 6d).”

The section about "inhibitors of the Ub-proteasome system" was moved to a new paragraph accordingly.